# Evaluation of aerosol- and gas-phase tracers for identification of transported biomass burning emissions in an industrially influenced location in Texas, USA

Sujan Shrestha[1], Shan Zhou[2,3], Manisha Mehra[1], Meghan Guagenti[1], Subin Yoon[2], Sergio L. Alvarez[2], Fangzhou Guo[2,3,a], Chun-Ying Chao[3], James H. Flynn III[2], Yuxuan Wang[2], Robert J. Griffin[3,4], Sascha Usenko[1], Rebecca J. Sheesley[1*]

[1]Department of Environmental Science, Baylor University, Waco, TX, USA

[2]Department of Earth and Atmospheric Sciences, University of Houston, Houston, TX, USA

[3]Depertment of Civil and Environmental Engineering, Rice University, TX, USA

[4]School of Engineering, Computing, and Construction Management, Roger Williams University, Bristol, RI, USA

[a]now at Aerodyne Research Inc., Billerica, MA, USA

Correspondence to: Rebecca J. Sheesley (Rebecca_Sheesley@baylor.edu)

## Abstract

As criteria pollutants from anthropogenic emissions have declined in the US in the last two decades, biomass burning (BB) emissions are becoming more important for urban air quality. Tracking the transported BB emissions and their impacts is challenging, especially in areas that are also burdened by anthropogenic sources like the Texas Gulf coast. During the Corpus Christi and San Antonio (CCSA) field campaign in Spring 2021, two long-range transport BB events (BB1 and BB2) were identified. The observed patterns of absorption Ångström Exponent (AAE), high-resolution time-of-flight aerosol mass spectrometer (HR-ToF-AMS) BB tracer ($f_{60}$), equivalent black carbon (eBC), acetonitrile and carbon monoxide (CO) during BB1 and BB2 indicated differences in the mixing of transported BB plumes with local anthropogenic sources. The combined information from HYSPLIT backward trajectory (BTs) and satellite observations revealed that BB1 had mixed influence of transported smoke plumes from fires in Central Mexico, the Yucatan peninsula, and the Central US, whereas BB2 was influenced majorly by fires in the Central US. The estimated transport time of smoke from the Mexican fires and the Central US fires to our study site were not too different (48-54 hours and 24-36 hours, respectively) and both events appeared to have undergone similar levels of atmospheric processing, as evident in the elemental ratios of bulk organic aerosol (OA). We observed an ageing trend for $f_{44}$ vs. $f_{60}$ and $f_{44}$ vs $f_{43}$ as a function of time during BB2, but not during BB1. Positive matrix factorization (PMF) analysis of OA showed that BB1 had a mixture of organics from aged BB emission with an anthropogenic marine signal while the oxidized organic compounds from aged BB emissions dominated the aerosols during BB2. The size distribution of aerosol composition revealed distinct characteristics between BB1 and BB2, where BB1 was found to be externally mixed, exhibiting a combination of BB and anthropogenic marine aerosols. On the other hand, BB2 exhibited internal mixing, ubiquitously dominated by aged BB aerosol. Our analysis from mobile and stationary

measurements highlights that both CO and acetonitrile are likely impacted by local sources even during the BB events and specifically that acetonitrile cannot be used as a unique BB tracer for dilute BB plumes in an industrially

influenced location. A suitable VOC tracer would need to be emitted in high concentrations during BB, resistant to degradation during transport, unique to BB and able to be measured in the field. This study does effectively demonstrate that AAE and aerosol BB tracers served as precise and effective tracers in these complex emission scenarios. Network deployment of multiwavelength photometers holds promise for enhancing our understanding of BB impacts on air quality and supporting informed decision-making for effective mitigation strategies in locations

with mixed sources and influence of dilute BB plumes. To demonstrate the relevance of such an aerosol optical network, we provide evidence of the potential regional impacts of these transported BB events on urban $O_3$ levels using measurements from the surface air quality monitoring network in Texas.

## 1. Introduction

Biomass burning (BB) activities emit fine particulate matter (PM$_{2.5}$, aerodynamic diameter smaller than 2.5 μm), volatile organic compounds (VOCs) and trace gases into the atmosphere. BB plumes can be transported across long distances and impact air quality in downwind locations (Rogers et al., 2020; Sciare et al., 2008; Sakamoto et al., 2015; Streets et al., 2003; Zhang et al., 2012; Morris et al., 2006; Markowicz et al., 2016; Forster et al., 2001). During long-range transport, the physical properties and chemical composition of the plume can be altered significantly by both plume ageing and dilution due to boundary layer mixing (Reid et al., 2005; Hung et al., 2020; Hodshire et al., 2019). In urban locations that are burdened by local anthropogenic sources, it is challenging to characterize and quantify the impacts of aged and/or dilute BB smoke plumes (Bein et al., 2008; Singh et al., 2012). Several approaches have been established to determine the impact of transported BB smoke on ambient air quality of downwind locations. This includes laboratory, field-based and satellite observations of aerosol composition and optical properties (de Gouw and Jimenez, 2009; Laing et al., 2016; Li et al., 2020; Zauscher et al., 2013; Zhou et al., 2017), and analyzing chemical and organic molecular markers of BB emissions (including non-sea salt potassium, acetonitrile and levoglucosan) (Yokelson et al., 2009; Bhattarai et al., 2019; Huangfu et al., 2021; Bond and Bergstrom, 2006; Mehra et al., 2019). Studies based on aerosol optical properties utilize wavelength dependence of aerosol absorption and scattering to identify aerosol type i.e., differentiate between black carbon (BC) from fossil fuel combustion, brown carbon (BrC) from BB and minerals from dust (Schmeisser et al., 2017). Absorption Ångström exponent (AAE) and scattering Ångström exponent (SAE) are commonly used intensive parameters to characterize the aerosol wavelength dependence (Bergstrom et al., 2007; Russell et al., 2010; Gyawali et al., 2009; Kirchstetter et al., 2004). In order for molecular or chemical markers to be used in identifying BB contribution, these markers must be conserved during atmospheric chemical reactions during the transport (Fraser and Lakshmanan, 2000). Further, for these markers to be detectable in urban locations, the specific tracer must be unique to BB emission and emitted in large quantity so that the compound is quantifiable above the urban background concentration.

The frequency, duration and burned area during wildfires in the Northwestern US increased over the last two decades under the changing climatic conditions (Westerling and Bryant, 2008; Westerling et al., 2006; Kasischke and Turetsky, 2006), implying an increase in the concentration of air pollutants during wildfire seasons (i.e., spring and summer) as a result of these fires. These impacts can be observed on a regional scale (Jaffe et al., 2008). For example, several studies have shown that the transported pollutants from BB emissions in the Alaska, Canada and Northwestern US can exacerbate ozone (O$_3$), CO, BC and PM$_{2.5}$ levels in Houston, Texas for several days (Lei et al., 2018; McMillan et al., 2010; Morris et al., 2006; Schade et al., 2011). Wildfires and agricultural burning in the Central Mexico and the Yucatan peninsula peak during the spring-summer season and also transport pollutants to the Southern US (Wang et al., 2018; Rogers and Bowman, 2001; Yokelson et al., 2013). Previous studies have documented emissions of trace gases, VOCs and particulates, and evolution of O$_3$ from forest fires and agricultural burnings in the southeastern US (Müller et al., 2016; Liu et al., 2016). It has not been reported whether fires in these regions are also increasing. Jaffe and Wigder (2012) conducted a comprehensive review of various factors that contribute to O$_3$ production from wildfire

emissions. These factors included emissions of $O_3$ precursors ($NO_x$ and VOCs), combustion efficiency, photochemical reactions, the influence of aerosols on chemistry and radiation, as well as local and downwind meteorological patterns.

On the contrary, literature have also reported carbonaceous aerosols and organics in the BB plumes can absorb and scatter incoming solar radiation, and reduce the photolysis of atmospheric trace gases, thereby reducing the surface $O_3$ production (Jiang et al., 2012; He and Carmichael, 1999; Tang et al., 2003). Thus, the interaction between meteorology and chemistry of the BB plume plays a crucial role in governing the effects on surface $O_3$ in the downwind regions.

Wang et al. (2018) have shown that the transport of Central Mexican and Yucatan BB emissions adversely impacted surface air quality at several major urban centers along the Gulf Coast, including Houston and Corpus Christi in Texas. The episodic transport events of BB emissions can result in $O_3$ and $PM_{2.5}$ exceedances of the air quality standards across several metropolitan areas in Texas. The Texas Commission on Environment Quality (TCEQ) operates a network of surface air quality monitoring stations in Texas, but the measurements are largely limited to criteria

pollutants. Realtime observational data integrated with satellite observations and transport models may improve efforts to track the transported BB emissions, locate the source regions, understand the plume ageing, and analyze its impacts on surface air quality.

Although Texas is the second-most populous state in the US, with multiple industrial and economic urban centers, many of the previous air quality studies focused on the Houston-Galveston-Brazoria and Dallas-Fort Worth areas

(Parrish et al., 2009; McMillan et al., 2010; Yoon et al., 2021; Anderson et al., 2019; Shrestha et al., 2022; Guo et al., 2021). To better understand air quality drivers in emerging Texas cities, the San Antonio Field Study (SAFS) 2017 investigated ambient concentration and sources of VOCs and trace gases as well as physical and chemical processes that control $O_3$ (Guo et al., 2021; Shrestha et al., 2022; Anderson et al., 2019). Results from the SAFS 2017 study highlighted the need to characterize the influence of upwind sources and long-range transport on air quality in San

Antonio. To address these outstanding questions from SAFS 2017, the Corpus Christi and San Antonio (CCSA) Field Study was conducted in spring 2021 (Zhou et al., 2023). Corpus Christi is upwind of San Antonio when the wind is coming from the south-southeast. Historical wind data analysis reveals that during the spring months in San Antonio, the prevailing wind direction is predominantly southeasterly (Guo et al., 2021). This mobile and stationary field experiment was designed to measure the impact of local emissions and transported pollution on air quality in Corpus

Christi and San Antonio. This manuscript primarily focuses on BB transport events identified during stationary measurement at Port Aransas (PA), a Gulf Coast city near Corpus Christi, during the field campaign (see Fig. 1). The goals were to (i) study the physical and chemical properties of transported BB smoke and their impact on background air quality in PA, (ii) identify fire source regions and understand transport times, ageing and dilution of smoke plumes and (iii) evaluate challenges of using BB tracers in an industrialized area like PA. Finally, we included general

comments on extending permanent in-situ monitoring networks with low-cost aerosol optical measurements for identifying BB events are offered.

## 2.    Method:

### 2.1.    Site description

The stationary measurements were performed at a beachfront site in PA, TX (27.803°N, 97.077°W) from April 3 – 15, 2021. The sampling site is approximately 4 km southwest of the mouth of the Corpus Christi Ship Channel into the Gulf of Mexico and 35 km directly east of Corpus Christi's urban core. Oil and gas wells lie in every direction from the study site (green star) (Fig. 1a). The instrumentation was housed in a Baylor University/ University of Houston/ Rice University-operated mobile air quality lab (MAQL2). MAQL2 is a 35-m³ insulated air-conditioned trailer with a ~9-m telescoping tower and inlet box that extends above the trailer during stationary measurements (Fig. S1). The aerosol inlet has a PM₂.₅ cyclone and stainless-steel bellows. The inlet lines inside the trailer were made as short as possible (0.5 m); these lengths of tubing were insulated to minimize wall loss and vaporization effects associated with temperature changes between the ambient air and inside the trailer. A heated sampling line set at 70 °C, manufactured by Atmos-Seal Engineering Inc., was used for VOC measurements. During mobile measurements, the MAQL2 was towed by a Ford F-250 truck with an air-ride system installed to minimize the vibration during motion and a generator was carried in the bed of the truck to provide electrical power. The inlet box was positioned above the front bumper of the vehicle, forward of the generator and truck exhaust, to avoid self-sampling during mobile measurements (Fig. S1).

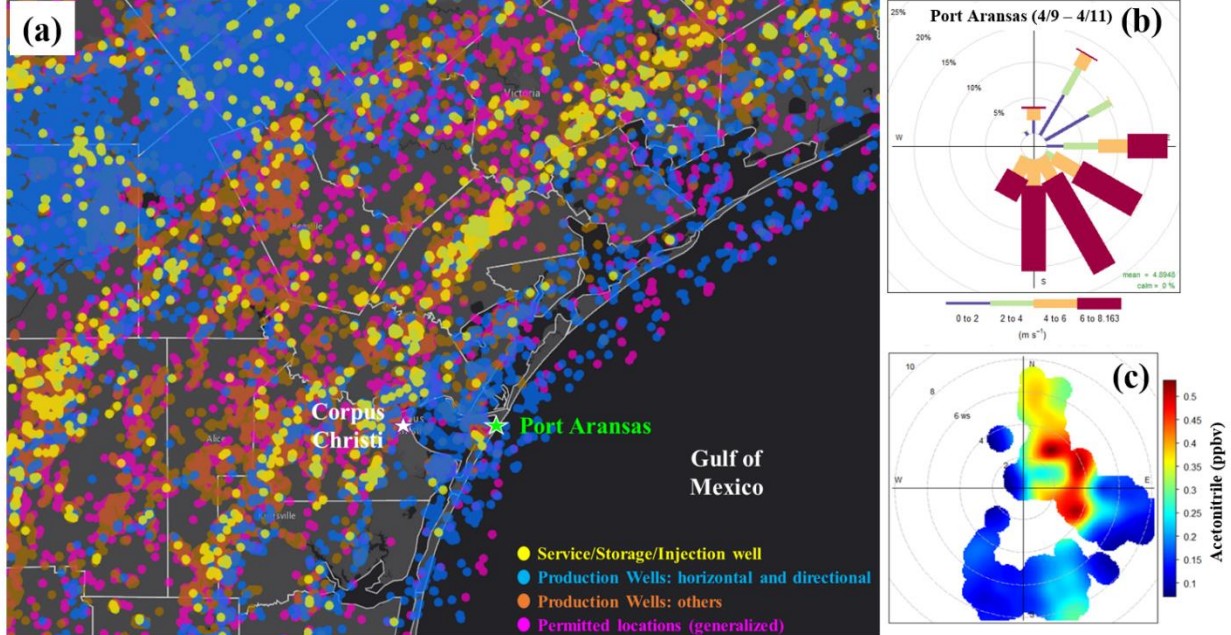

**Figure 1. (a)** GIS map of oil and gas activities in Texas (study site, PA is shown by green star), **(b)** wind rose for PA and **(c)** pollution rose plot for acetonitrile during the period of interest (4/9/2021 – 4/11/2021). The GIS map in **panel a** was obtained from maps.fractracker.org (latest as of 05/08/2021).

## 2.2. Instrumentation

### 2.2.1. Aerosol optical parameters

The aerosol light absorption coefficient ($\sigma_{abs}$) was measured with a 3λ tricolor absorption photometer (TAP, Brechtel Inc., Hayward, CA) at wavelengths of 365, 520, and 640 nm. The TAP is the commercially available version of the National Oceanic and Atmospheric Administration (NOAA's) continuous light absorption photometer (CLAP) (Ogren et al., 2017). The TAP consecutively samples through eight sample filter spots and two reference filter spots. During deployment at PA, the TAP was set to rotate to the next filter spot when a filter spot's transmission reached 50 %.

The light scattering coefficient ($\sigma_{scat}$) was measured using an integrating nephelometer (model 3563, TSI Inc., Shoreview, MN) at wavelengths of 450, 550, and 700 nm. During the campaign, the TSI nephelometer was calibrated against zero air and carbon dioxide ($CO_2$) (Anderson and Ogren, 1998). The measured values were corrected for angular scattering and truncation error (Anderson and Ogren, 1998; Bond et al., 2009) using the relationship: $\sigma_{corrected}$ = correction factor (C) × $\sigma_{neph}$ where C is the correction factor, $\sigma_{neph}$ is the scattering coefficient reported by the instrument, and $\sigma_{corrected}$ is the corrected scattering coefficient (Shrestha et al., 2018). The correction factor (C) was calculated using Eq. (1) where the values for constants a and b were obtained from Anderson and Ogren (1998) and SAE was calculated from the scattering coefficients measured during this study.

$$C = a + b * SAE^{\frac{\lambda 1}{\lambda 2}} \tag{1}$$

Using 5-min averages, AAE and SAE were calculated as the negative slope of the linear fit of the optical parameter versus wavelength on a log-log plot (Bergstrom et al., 2007; Bond and Bergstrom, 2006; Kirchstetter et al., 2004). AAE and SAE provide information about the wavelength-dependence of absorption and scattering, respectively (Schmeisser et al., 2017). Generally, AAE values of approximately 1 characterize fresh BC, whereas, BrC and mineral oxides show strong preferential light absorption in the UV range resulting in an enhancement in AAE values with respect to BC (Bond and Bergstrom, 2006; Bergstrom et al., 2007). SAE values are inversely related to the particle size distribution within the measured sample, so that generally SAE values less than 1 indicate size distribution dominated by coarser particles while those greater than 1 indicate that finer particles dominated the scattering aerosol (Schuster et al., 2006). The narrow spectral range of TAP (365nm - 640 nm) compared to other aerosol absorption measurements like aethalometer (AE33) may result in lower range of AAE. Therefore, in this study, AAE above 1.2 (i.e., average AAE during non-BB influenced period + two times standard deviation) is used to identify events that lie above the baseline AAE for a given site (discussed in Section 3.3.1), rather than absolute AAE value from the literature. While the variability in aerosol optical measurements between different instruments has been extensively studied in previous literature (Laing et al., 2020; Ogren et al., 2017; Ogren, 2010; Bond et al., 1999), it is not the primary focus of this manuscript. Previous intercomparison study has demonstrated excellent agreement between long-term measurements between the CLAP (NOAA's version of TAP) and the particle soot absorption photometer (PSAP) at multiple sites (Ogren et al., 2017), and have indicated that TAP and AE33 intercompare when a different correction factor is applied to the AE33 absorption coefficient (Laing et al., 2020). The absorption coefficient data from this study is available (link below in the data availability section). This will enable future studies to access and utilize data from this study for the investigation and comparison with other instruments that use different protocols for BC calculation.

Single Scattering Albedo (SSA) is the ratio of $\sigma_{scat}$ to extinction coefficient ($\sigma_{scat} + \sigma_{abs}$), which provides information about the top-of-atmosphere forcing due to aerosol, i.e., absorbing, or scattering nature of the sampled aerosol. An

SSA value greater than 0.95 represents aerosol with a net cooling effect, while a value less than 0.85 will result in net warming. The SSA values between 0.85 and 0.95 may represent warming or cooling effect depending upon surface albedo and cloud cover (Ramanathan et al., 2001). The wavelengths for $\sigma_{scat}$ and $\sigma_{abs}$ were not the same, so to calculate SSA at 550 nm, the $\sigma_{abs}$ measured at 540 nm was converted to that applicable to 550 nm using equation below:

$\qquad \sigma_{abs}^{550} = \sigma_{abs}^{540} \times \left(\frac{\lambda_{540}}{\lambda_{550}}\right)^{AAE_{365-640}}$ (2)

### 2.2.2. PM$_{2.5}$ filter sampling

PM$_{2.5}$ samples were collected on 90-mm diameter quartz fiber filters (Pall Corporation, Port Washington, NY, USA) using a medium-volume (90 L min$^{-1}$; URG Corporation, Chapel Hill, North Carolina, USA) sampler at the Texas A&M Corpus Christi campus. Detailed discussion regarding filter collection protocols is reported in Yoon et al.

(2021). We calculated equivalent black carbon (eBC) mass concentrations from the absorption coefficient measured by TAP at a wavelength of 520 nm using mass absorption cross-section (MAC) values determined from the PM$_{2.5}$ filter samples. Details about the eBC calculation are presented in the *Supplementary Section S1*. Using the method discussed in the *Supplementary Section S1*, the derived MAC at 520 nm was 11.45 m$^2$g$^{-1}$.

### 2.2.3. Real-time, size-resolved aerosol composition

An Aerodyne (Billerica, MA, USA) HR-ToF-AMS was used for size-resolved chemical characterization of non-refractory submicron aerosols (NR-PM$_1$) (DeCarlo et al., 2006). Detailed discussion regarding HR-ToF-AMS operation and data handling followed during this study can be found in our previous publication (Zhou et al., 2023). In brief, the size-resolved NR-PM$_1$ mass concentration and chemical composition were analyzed using the standard HR-ToF-AMS data analysis toolkit (SQUIRREL v1.64 and PIKA v1.24). Table S2 listed the MDLs of the five HR-

ToF-AMS species (organic, sulfate, nitrate, ammonium, and chloride). Positive matrix factorization (PMF) analysis on the combined spectral matrices of organic and inorganic species of the HR-ToF-AMS (Zhou et al., 2017; Paatero and Tapper, 1994) identified seven organic aerosol (OA) factors associated with distinct sources and chemical and physical properties, which includes i) hydrocarbon-like OA (HOA) associated with traffic emissions, ii) biomass burning OA (BBOA), iii) less-oxidized oxygenated OA (LO-OOA) likely representing fresher secondary OA (SOA),

iv) more-oxidized OOA (MO-OOA) likely representing more aged and processed SOA, v) less oxidized OOA that was associated with ammonium nitrate (AN-OOA), vi) highly oxidized OOA associated with ammonium sulfate (AS-OOA), and vii) highly oxidized OOA associated with acidic sulfate (acidic-OOA). Details on the PMF analysis method and results evaluation can be found in the *Supplementary Section S2*. Further, the f$_{60}$ value (i.e., the fraction of the signal at *m/z* 60 (mostly C$_2$H$_4$O$_2^+$) in the OA spectrum) above 0.3 % were used as markers for BB emissions (Docherty

et al., 2008; Cubison et al., 2011).

### 2.2.4. Trace Gases and meteorological data

Nitric oxide (NO), nitrogen dioxide ($NO_2$), nitrogen oxides ($NO_x = NO + NO_2$), total reactive nitrogen ($NO_y$), CO and $O_3$ were measured during the campaign. $O_3$ measurements were conducted using a modified Thermo Environmental, Inc., Model 42C instrument, which utilizes chemiluminescence (CL) with NO gas to measure $O_3$. NO and $NO_2$ were
measured using CL instruments (Air Quality Design (Golden, CO)). The $NO_y$ was measured with a molybdenum oxide catalytic converter inlet and subsequent CL $NO_x$ analyzer. The CO was measured using off-axis integrated cavity output spectroscopy (Los Gatos Research, Inc., Li-7000). Greater detail about trace gas measurements are presented in our previous publications (Shrestha et al., 2022; Guo et al., 2021). The MDL and uncertainty for trace gas measurements during the campaign are presented in Table S3.

Basic meteorological parameters, including wind speed and direction, temperature, and relative humidity, were measured continuously using RM Young 86000 ultrasonic anemometer. Fig. 1b shows that southeast wind was dominant at PA with intermittent wind from other directions during the campaign.

**2.2.5. Volatile Organic Compounds (VOC)**

A quadrupole proton transfer reaction- mass spectrometer (PTR-MS Q300; Ionicon Analytik, Austria) was used to
measure VOCs during this study. In the PTR-MS, target gas molecules are ionized by proton transfer from protonated water ($H_3O^+$). The ionized material is then detected and quantified using a quadrupole mass spectrometer. A more detailed description of the PTR-MS is given in other studies (Lindinger and Jordan, 1998; de Gouw et al., 2003b; de Gouw and Warneke, 2007). A sample drying system similar to that used by Jobson and McCoskey (2010) was implemented to reduce any effects of water vapor that can occur with operating the PTR-MS at a lower E/N (100 Td).
Greater details regarding PTR-MS operation, calibration and VOC data analysis followed in this study are presented in our previous publication (Shrestha et al., 2022). The MDLs and uncertainty of the measured VOCs during the campaign are presented in Table S4.

**2.3. Satellite observations**
**2.3.1. Active fire count and AOD**

The ground-based measurements have been supported by the analysis of satellite aerosol optical depth (AOD) data obtained from the Moderate Resolution Imaging Spectroradiometer (MODIS), mounted onboard the Aqua and Terra satellites. The MODIS AOD gridded at a 10 x 10 km spatial was averaged for each day. This study uses level 3 AOD at 550 nm over land and ocean product for understanding trends in smoke aerosol loading (Remer et al., 2005; Levy et al., 2007).

Information about the daily active fires was obtained from the Visible Infrared Imaging Radiometer Suite (VIIRS) satellite observations. The VIIRS imagery-resolution bands sense 32, 375 m-pixel lines per scan with a field view of 112.56° (Li et al., 2020; Cao et al., 2014; Wolfe et al., 2013). The active fire confidence values below 70 % were eliminated during the data processing.

### 2.3.2. Satellite imagery of smoke plumes

The smoke map generated by NOAA Hazard Mapping System (HMS) was used to understand the spatial distribution of visible smoke plumes across North America. The NOAA HMS graphics system is an interactive satellite image developed by the National Environmental Satellite, Data, and Information Service (NESDIS). The satellite imagery can be downloaded from the NOAA smoke product website (https://satepsanone.nesdis.noaa.gov/FIRE/fire.html). These maps provide daily information on the horizontal distribution and density of the smoke plumes in the region (Rogers et al., 2020; Rolph et al., 2009; Fischer et al., 2018).

### 2.4. Backward trajectory analysis

The NOAA HYSPLIT model (Draxler and Hess, 1998; Stein et al., 2015) was used to simulate 72-h backward trajectories (BTs) at different starting heights (50, 100 and 500 m) every hour from April 9 through April 12, 2021 (CDT) at the PA site. The BTs at all three starting heights reported similar results; therefore, we chose the 50-m starting height for further analysis (Fig. S4). The HYSPLIT model has been used extensively for atmospheric transport and dispersion research in the last three decades. In this study, the HYSPLIT model was used to study possible source regions and estimate the age of the airmasses arriving at the study site during the BB events. Meteorological data from the Global Data Assimilation System (GDAS) with $0.5° \times 0.5°$ spatial resolution were used in this study.

### 3. Results and discussion

During the stationary period when the MAQL2 was deployed in PA for the CCSA study, potential BB events were identified, first through the daily NOAA HMS updates of smoke across the Gulf and in the greater Corpus Christi area and then through evaluation of in-situ measurements of aerosols, VOCs, and trace gas from the MAQL2. Two BB events were identified on April 10 (11:00 – 23:00 CST) and April 11 (6:45 – 14:00 CST) at PA and are referred to as BB1 (orange shade) and BB2 (pink shade) hereafter (Fig. 2). The two BB events were first distinguished based on the observed pattern of enhancement in AAE and the HR-ToF-AMS tracer, $f_{60}$ (*Section 3.1*); the accuracy of the AAE identification of BB influence was assessed in comparison with $f_{60}$. To better understand transport times and potential plume age, we analyzed possible source regions using BTs and satellite observations *(Section 3.2)*. Based on these results, we considered aerosol chemical speciation of NR-PM$_1$ (*Section 3.3*) and evaluated the efficacy of gas-phase BB tracers (including CO and acetonitrile) in an industrialized urban environment (*Sections 3.4 and 3.5*). Finally, we discuss the potential implications of this BB event on Texas urban air quality.

### 3.1. Identifying biomass burning using aerosol optical properties

The aerosol optical, aerosol chemical speciation, trace gases and meteorological measurements from PA highlight changes in composition during the April 9 - 11 period of interest; this includes the day prior to the identified BB event.

Based on the direction of the surface wind, the in-situ measurements were separated into marine and continental periods, while the BB designation was defined by the aerosol indicators (AAE and $f_{60}$). The measurement statistics during BB1, BB2, marine background and continental airmass periods are presented in Table 1. The short-duration events associated with local combustion that impact AAE and $f_{60}$ values were removed from the marine average (see Fig. 2). Note that the statistics presented in this study are for a short period of interest (April 9 - 11) within a total campaign (April 3 -15), so the averages presented here differ from campaign averages reported in Zhou et al. (2023). Unless otherwise specified, all the data presented in this study pertains to the period of interest, i.e., April 9 - 11.

The $\sigma_{abs}$ values for the ultraviolet-visible range (365-640 nm) were significantly higher during BB1 and BB2 (e.g., $5.57 \pm 2.56$ and $6.89 \pm 2.42$ Mm$^{-1}$, respectively, at 520 nm) compared to marine and continental airmasses ($2.79 \pm 1.16$ and $4.12 \pm 1.31$ Mm$^{-1}$, respectively, at 520 nm) in the same week. However, the average $\sigma_{scat}$ in all three wavelengths (450 nm, 550 nm and 700 nm) at PA were similar between BB and background marine airmass (Table 1). This does not agree with the studies conducted at remote locations and during airborne measurements of relatively fresh plumes that reported enhancement in aerosol scattering, mass concentration and number concentration during atmospheric transport of BB aerosols (Laing et al., 2020; Yokelson et al., 2009; Hobbs et al., 2003). It is interesting that the mean background $\sigma_{scat}$ and $\sigma_{abs}$ (56.52 Mm$^{-1}$ and 2.79 Mm$^{-1}$) at PA were higher than some of the other coastal locations in the US such as Trinidad Head ($\sigma_{scat} = 21.51$ Mm$^{-1}$ and $\sigma_{abs} = 0.94$ Mm$^{-1}$) and Pt. Reyes in California ($\sigma_{scat} = 40$ Mm$^{-1}$ and $\sigma_{abs} = 0.69$ Mm$^{-1}$) and Cape Cod in Massachusetts ($\sigma_{scat} = 16.08$ Mm$^{-1}$ and $\sigma_{abs} = 1.10$ Mm$^{-1}$) (Oltmans et al., 2008; Berkowitz et al., 2005; Titos et al., 2014). The higher background $\sigma_{scat}$ and $\sigma_{abs}$ at PA demonstrate the influence of anthropogenic emissions including shipping activities and oil and gas extraction on background aerosol in the Gulf of Mexico (Zhou et al., 2023).

The enhancement in $\sigma_{abs}$ during the BB events was higher in the UV wavelength compared to longer wavelengths (Table 1). This wavelength dependency in aerosol absorption resulted in high AAE during BB1 and BB2 ($1.2 \pm 0.2$ and $1.3 \pm 0.2$, respectively) compared to the marine ($0.66 \pm 0.14$) and continental airmasses ($0.94 \pm 0.24$) (Fig. 2a). The marine airmasses had AAE significantly less than 1, similar to other coastal locations such as Graciosa Island in Azores, Portugal (average AAE of 0.65) (Jefferson, 2010) and Pt. Reyes in California, USA (average of ~0.5) (Berkowitz et al., 2005; Schmeisser et al., 2017). The continental airmasses at PA during this study had AAE ~1, which is routinely reported in urban aerosols that contain BC from fossil fuel combustion. Ambient aerosols impacted by BB can include brown carbon (BrC), which preferentially absorbs at lower wavelengths, resulting in an increased AAE (in excess of 2) (Bergstrom et al., 2007; Bond and Bergstrom, 2006; Kirchstetter et al., 2004). However, laboratory and field-based studies have reported a wide range of AAE values for different biomass fuel and burn conditions (0.55 to more than 3) (Gyawali et al., 2009; Bahadur et al., 2012; Pokhrel et al., 2016; Kirchstetter et al., 2004). Additionally, as the BrC emitted during wildfires decays during atmospheric transport (Forrister et al., 2015; Liu et al., 2016), there may be a subsequent decrease in AAE of the plume aerosols due to photobleaching (Reid et al., 2005; Eck et al., 2001; O'Neill et al., 2002). Liu et al. (2021) reported that the organic aerosol absorptivity decreases significantly during evolution when there is light present. When considering the decrease during transport

and dilution associated with mixing with local aerosol, it is not unexpected that the AAE during BB1 and BB2 at PA showed only a minor enhancement above the marine and continental backgrounds. In fact, similar AAE values have been reported for transported BB plumes impacting urban locations during South African (Bergstrom et al., 2007) and Yucatan fires (Marley et al., 2009). Although the AAE was impacted by the BB event, the SAE was consistent (1.5 - 1.7) during the period of interest. Locations that are influenced by coarse marine mode aerosols exhibit lower SAE

(less than 1) (Costabile et al., 2013; Pandolfi et al., 2018; Titos et al., 2014). However, the slightly higher range of SAE observed at PA indicated additional influence of local anthropogenic emissions besides marine influence at the sampling site (Zhou et al., 2023).

BB1 and BB2 events had clear synoptic peaks of eBC, OA and $f_{60}$ (Fig 2 b & c). Both the eBC and OA concentrations were significantly enhanced during BB1 and BB2 while the $f_{60}$ was $0.35 \pm 0.12$ % and $0.42 \pm 0.10$ % during BB1 and

BB2 (Table 1). An $f_{60}$ value above 0.3% indicates BB influence (Zhou et al., 2017). Thus, the aerosol BB tracers have good agreement with respect to the BB designation during the period of interest. However, the trace gas and eBC reveal a more complicated scenario. There was a good correlation for eBC with CO ($r^2 = 0.62$) and $f_{60}$ ($r^2 = 0.75$) during BB2 and poor correlation with CO ($r^2 = 0.23$) and $f_{60}$ ($r^2 = 0.27$) during BB1 (Figs. S2 & S3). Fig. 3 a-c shows that the patterns of AAE, $f_{60}$, eBC and CO were different during BB1 and BB2. The more specific BB tracers (AAE

and $f_{60}$) had a different temporal trend than the more general combustion tracers (eBC and CO), indicating that PA was influenced by more than one type of combustion plume during BB1 and BB2. The high variability of the eBC and CO concentrations during BB1 is possibly driven by mixed plumes from different sources. Further, CO had high peaks for a couple of hours (18:00 – 20:00 CST) during BB1 when the wind speed was very low ($\sim$1 m s$^{-1}$). We assume that the high CO during that period was contributed by local non-BB combustion sources as indicated by elevated

HOA concentrations (e.g., traffic or other primary combustion emissions), a lack of enhancement in acetonitrile concentration (discussed in *Section 3.4*) and poor correlations of CO with eBC and $f_{60}$.

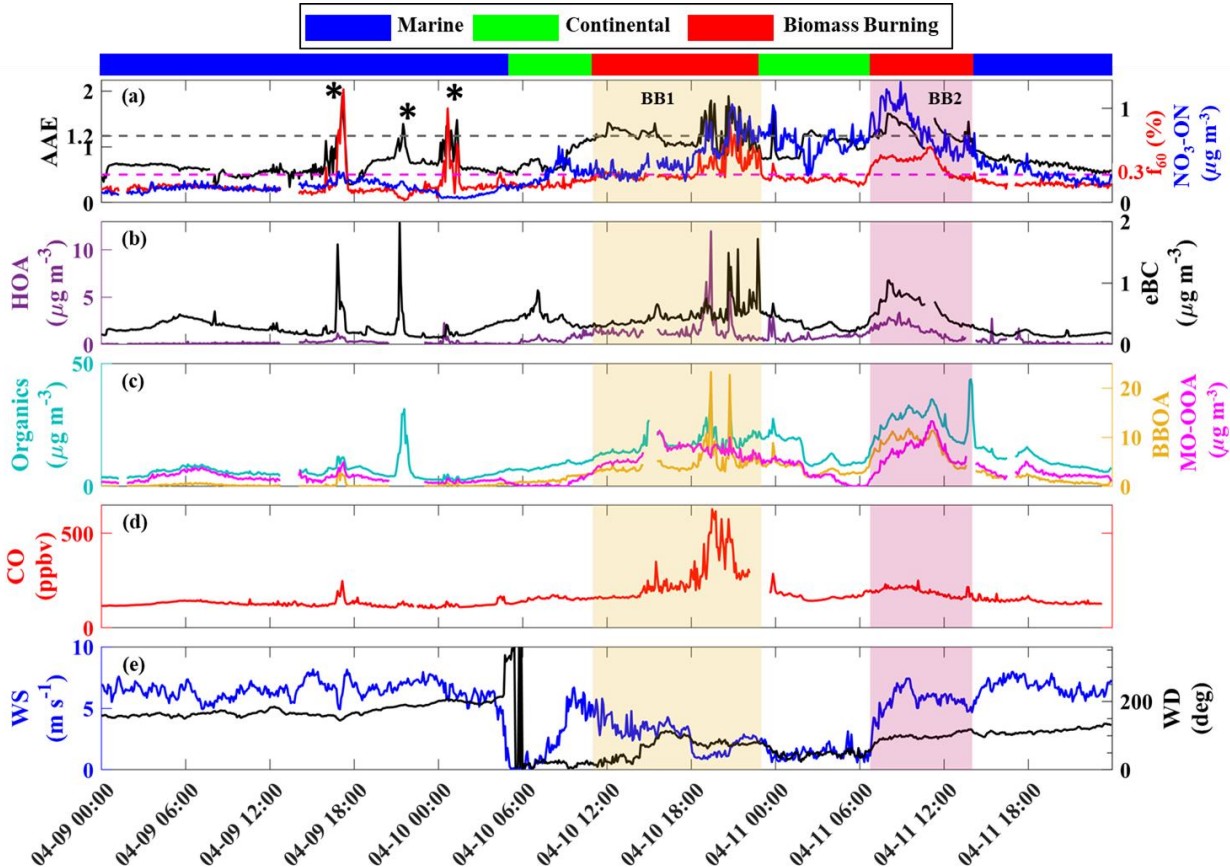

**Figure 2.** Time series plots for **(a)** AAE, $f_{60}$ (dashed lines in grey and magenta represents AAE = 1.2 and $f_{60}$ = 0.3 %, respectively), and the AMS measured nitrate signal attributed to organonitrates (NO$_3$-ON) **(b)** HR-ToF-AMS HOA
factor concentration, and eBC concentration, **(c)** OA concentration, and BBOA factor and MO-OOA factor concentrations **(d)** CO and **(e)** wind speed and direction. The short-duration events, indicated by the symbol * in **panel a**, associated with local combustion were removed from the marine average. The marine and continental airmass classifications are differentiated based on the surface wind direction measured during the campaign.

**Table 1:** Basic statistics (average ± standard deviation) of aerosol optical properties, aerosol speciation and select
gases during the period of interest (April 9 - 11). The short-duration events, indicated by the symbol * in Fig. 2a, associated with local combustion were removed from the marine average.

| Parameters | BB1 | BB2 | Marine | Continental |
|---|---|---|---|---|
| $\sigma_{abs}$ at 365 nm (Mm$^{-1}$) | 8.99 ± 4.52 | 11.5 ± 4.51 | 3.52 ± 1.49 | 5.81 ± 1.69 |
| $\sigma_{abs}$ at 520 nm (Mm$^{-1}$) | 5.57 ± 2.57 | 6.89 ± 2.42 | 2.79 ± 1.16 | 4.12 ± 1.31 |
| $\sigma_{abs}$ at 640 nm (Mm$^{-1}$) | 4.53 ± 2.05 | 5.53 ± 1.83 | 2.44 ± 1.02 | 3.49 ± 1.17 |
| $\sigma_{scat}$ at 450 nm (Mm$^{-1}$) | 64.9 ± 16.6 | 56.9 ± 12.7 | 73.3 ± 34.8 | 58.1 ± 36.2 |
| $\sigma_{scat}$ at 550 nm (Mm$^{-1}$) | 50.1 ± 15.0 | 41.2 ± 8.79 | 56.4 ± 26.8 | 43.2± 26.7 |
| $\sigma_{scat}$ at 700 nm (Mm$^{-1}$) | 34.2 ± 11.4 | 26.6 ± 5.34 | 37.4 ± 17.0 | 27.9 ± 16.9 |
| AAE | 1.21 ± 0.21 | 1.28 ± 0.16 | 0.66 ± 0.14 | 0.94 ± 0.24 |

| | | | | |
|---|---|---|---|---|
| SAE | 1.52 ± 0.19 | 1.71 ± 0.10 | 1.55 ± 0.34 | 1.62 ± 0.19 |
| SSA (550nm) | 0.90 ± 0.02 | 0.87 ± 0.03 | 0.95 ± 0.03 | 0.90 ± 0.05 |
| eBC (ng/m$^3$) | 487 ± 224 | 601 ± 211 | 243 ± 102 | 359 ± 114 |
| $f_{60}$ (%) | 0.35 ± 0.12 | 0.42 ± 0.10 | 0.17 ± 0.04 | 0.23 ± 0.04 |
| OA (µg/m$^3$) | 17.8 ± 4.81 | 26.4 ± 6.08 | 7.31 ± 3.93 | 11.3 ± 4.64 |
| Acetonitrile (ppbv) | 0.42 ± 0.10 | 0.18 ± 0.05 | 0.19 ± 0.07 | 0.27 ± 0.09 |
| CO (ppbv) | 259 ± 117 | 192 ± 21.7 | 130 ± 14.4 | 162 ± 20.2 |

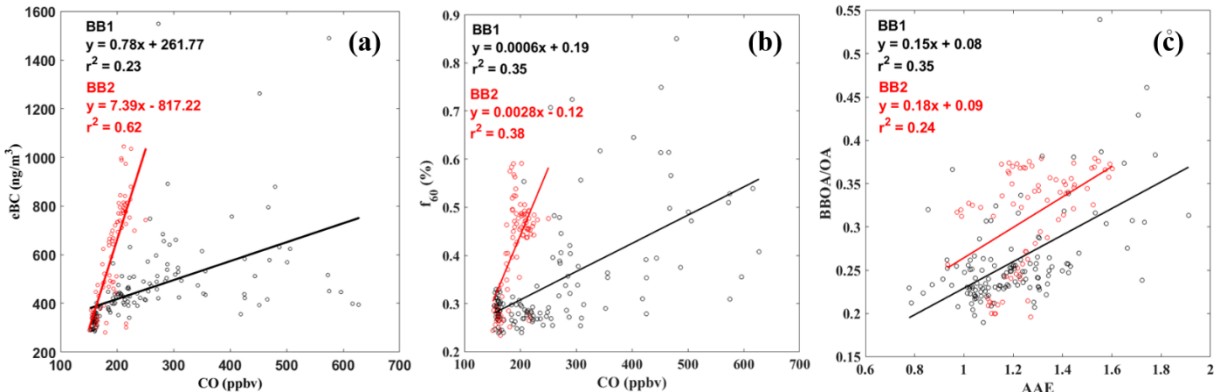

**Figure 3.** Correlation of (**a**) eBC versus CO (**b**) $f_{60}$ versus CO and (**c**) BBOA/OA ratio versus AAE during BB1 and BB2. The slope of the regression lines in **panel b** is close to zero due to the difference in the magnitude of the $f_{60}$ value and CO concentration.

### 3.2. Analysis of potential biomass burning source regions using satellite data and backward trajectory analysis

For this study, we evaluated MODIS AOD, VIIRS fire count, NOAA HMS smoke product, and HYSPLIT BTs to identify the fire source regions and estimate the plume age/transport times of BB1 and BB2, following a methodology similar to previous studies (Laing et al., 2016; Zhou et al., 2017; Deng et al., 2008; Mathur, 2008). The NOAA HMS smoke product indicated that PA had smoke in the column from April 10 - 16 (Fig. 4); however, it does not provide information about vertical distribution and boundary layer mixing of the smoke (Jaffe et al., 2020; Buysse et al., 2019). The influence of BB on surface air quality was evident in the ground-based observations only on April 10 and 11 (see *Section 3.1*). Because we are most interested in the days with surface air quality impacts, we focused on source regions and transport during April 10 - 11. Extensive fire detects were evident in Central Mexico, the Yucatan peninsula, and the Central US during this period (Fig. 4). BTs ending at PA shifted gradually from Mexico to the Northern US over the course of the day on April 10 and remained from the Northern US through April 11 (Fig. S4), intersecting with dense fire hotspots in the Central US (Oklahoma and Kansas). The air parcel heights during BB1 and BB2 were generally below 1500 meters above ground level (m.a.g.l.) when they passed over the active fire locations in the Central US (Fig. 4). As discussed above, the HMS smoke product indicated a smoke plume extending from Mexico,

over the Gulf of Mexico, and to PA on both days during BB1 and BB2, indicating the possibility of influence of the fires in the Mexican region, which complicates the assignment of a specific source region. The smoke cover in the Gulf of Mexico resulted in higher AOD values ($0.5 \pm 0.1$; white square region in Fig. 4) on April 10 and 11 compared to the days prior to the event ($0.3 \pm 0.1$), indicating heavy loading of aerosols on those days. Specifically, the MODIS

grid that includes PA reported higher AOD during BB events (0.24) compared to the days prior to the event (0.18). Therefore, based on the BTs and satellite analysis, we assume that BB1 had mixed influence of transported smoke plumes from fires in Central Mexico, the Yucatan peninsula, and the Central US, whereas BB2 was influenced predominantly by fires in the Central US. Based on the combined information of fire hotspots and the BTs, we estimate that the transport time of smoke from the Mexican fires and the Central US fires ranged from 48 - 54 hours and 24 -

36 hours, respectively, before arriving at PA. The difference between the two BB events that was evident in the BB tracer analysis is then supported by differences in the BTs and source region analysis. The apparent local influence on CO and HOA is not specifically addressed by the satellite and BT analysis except to confirm that there were no local BB sources immediately upwind of PA at this time. In a broad sense, these results highlight the importance of integrating ground-based monitors, including permanent in-situ air quality monitoring networks and intensive

deployments, and satellite observations to understand the impact of smoke on surface air quality.

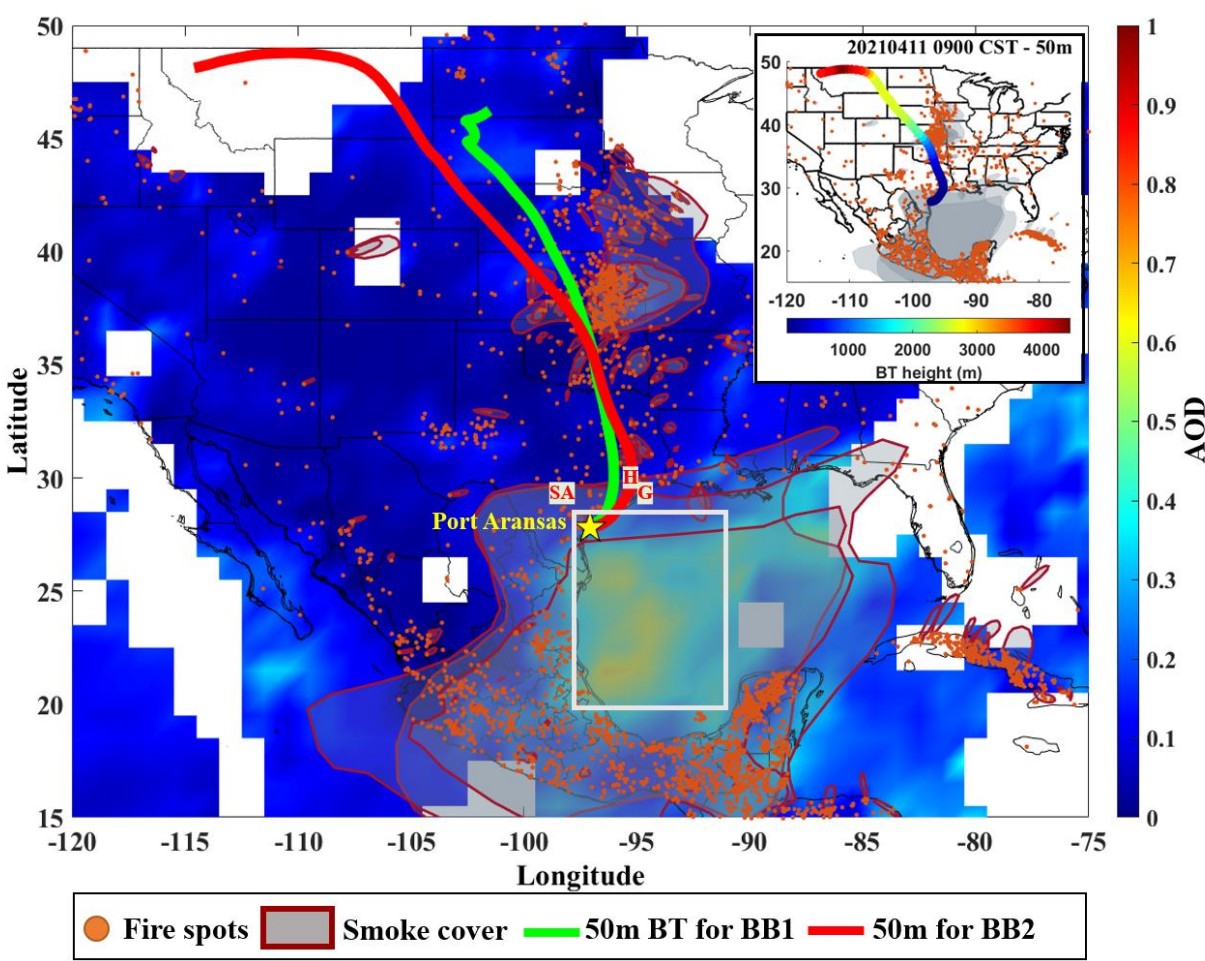

**Figure 4.** Spatial distribution of average AOD from Aqua and Terra satellites (April 10 – 11, 2021). The white outlined box shows the grid size considered for calculating the average AOD. VIIRS active fire, NOAA HMS smoke and BTs are included in the main map and the inset. The inset plot also includes trajectory heights. The end times of the BTs are chosen to represent the middle of the BB1 (4/10/21 20:00 CST) and BB2 (4/11/21 09:00 CST) observed in the ground-based measurement. The study site PA is denoted by a yellow star symbol. H, G and SA represents geolocations of other major cities in Texas (Houston, Galveston, and San Antonio, respectively). This map was created in MATLAB.

### 3.3. Aerosol chemical composition during biomass burning events

To further characterize the two BB events at PA, we assessed the chemical speciation and the particle size-based OA composition from the HR-ToF-AMS. Figures 5 and S5 provide an overview of the aerosol composition from April 9 - 11. Differences in composition among marine background, continental and the two BB events are used here both to validate the BB designation and to further characterize the BB plume.

Previous studies have shown a significant increase in $NR\text{-}PM_1$ concentration during BB events when the plumes are relatively fresh and the sampling locations do not have immediate anthropogenic sources (Zhou et al., 2017; Hu et al., 2016). In contrast, the average $NR\text{-}PM_1$ concentrations for this study were similar amongst BB1 ($27.84 \pm 6.53$ µg/m$^3$), BB2 ($33.96 \pm 6.53$ µg/m$^3$), marine background ($27.59 \pm 8.82$ µg/m$^3$) and continental airmass ($24.17 \pm 9.42$ µg/m$^3$). This result is consistent with the scattering coefficient measurements discussed in *Section 3.1*. These observations indicated that marine airmasses at PA were highly polluted with a submicron aerosol loading that included contributions from anthropogenic activities in the Gulf of Mexico (Zhou et al., 2023) and that the dilute BB plumes did not enhance the absolute aerosol concentration at the surface. Although the total $NR\text{-}PM_1$ concentrations were similar, the aerosol compositions changed drastically among the BB events, marine background and continental airmasses. During the BB events, organics dominated the aerosol composition (66 % and 78 % of $NR\text{-}PM_1$ during BB1 and BB2, respectively). The OA fraction was enhanced during BB1 and BB2 compared to the background marine (29 % of $NR\text{-}PM_1$) and continental airmasses (49 % of $NR\text{-}PM_1$). The OA fraction in the BB depends on the fuel burned and stage of fire (i.e., smoldering and flaming), and evolution during the transport. Generally, high values of OA fraction (greater than 90 %) have been reported for forest fires in the Amazon (Artaxo et al., 2013), North America (Kondo et al., 2011; Zhou et al., 2017) and Africa (Capes et al., 2008). Slightly lower OA fractions (~60-70 %) have been reported in Asia (Chakraborty et al., 2015; Kondo et al., 2011). Notably, the OA fractions observed during the BB events during this study are comparable to that reported for the Yucatan fires ($60 \pm 11$ %) (Yokelson et al., 2009). We assume that the moderate range of OA fraction in this study compared to the freshly emitted BB emissions is due to boundary layer mixing of the BB plume with local emissions and ageing during transport, as evidenced by the $NR\text{-}PM_1$ composition.

The background marine $NR\text{-}PM_1$ was dominated by sulfate ($SO_4^{2-}$; 58 % of the total $NR\text{-}PM_1$ mass) whereas the $SO_4^{2-}$ fraction was second most prevalent constituent of continental airmass (38 % of the total $NR\text{-}PM_1$ mass), just slightly lower than the organic fraction (46 % of the total $NR\text{-}PM_1$ mass). The $SO_4^{2-}$ fraction of $NR\text{-}PM_1$ was significantly

lower during BB1 (23.07 ± 3.96 %) and BB2 (14.27 ± 7.32 %) but was still higher than in previous studies from Yucatan (0.89 ± 0.56 %) and Amazon fires (1.95 ± 0.83 %) (Yokelson et al., 2009; Ferek et al., 1998). The forest fires in this region are not a significant source of sulfate aerosol (Collier et al., 2016; Zhou et al., 2017; Yokelson et al., 2009). Zhou et al. (2023) reported that anthropogenic sulfate remained the dominant sulfate source in the Gulf, coastal, and continental airmasses during the same study, whereas mass concentrations of sea-salt sulfate and biogenic sulfate were the largest in the Gulf airmasses and decreased with increasing continental influences. These results from Zhou et al. (2023) highlighted anthropogenic shipping emissions over the Gulf of Mexico as a major contributor to sulfate at PA during the campaign.

Figure S5 shows that the NR-PM$_1$ composition during the marine background period was dominated by sulfate, and the OA composition was dominated by AS-OOA while acidic-OOA was also elevated, suggesting influences from processed shipping emissions. During the BB periods, however, organics became the dominant NR-PM$_1$ component, and the mass fractional contribution of sulfate aerosol decreased dramatically while BBOA and MO-OOA increased. This suggests that the OA composition during the BB periods were predominantly driven by BB plumes; marine anthropogenic activities likely had minimal influence on the non-BB OAs.

The PMF analysis of NR-PM$_1$ can facilitate our characterization of the two different BB events (Figs. 2 and S5). Using both the specific tracers, AAE and $f_{60}$, in addition to the PMF results, we see interesting differences between BB1 and BB2. Although AAE, $f_{60}$, CO, and BBOA generally increased during BB1 and BB2, the period from 18:00 – 22:00 on April 10 did not show good agreement among the tracers. The HOA and BBOA factors correlated ($r^2$ = 0.79) during BB1 (Fig. S2), but the BBOA factor did not follow the same trend as the AAE and $f_{60}$ during this short time period. The HOA factor was mostly associated with fresh or local combustion sources and more closely mimicked the CO trends (discussed in *Section 3.1*) during the time period of BB1, which may again hint that the extreme CO peak in the evening of April 10 was a local combustion source not of BB origin. Interestingly, the MO-OOA factor mirrored some of the more gradual increases seen in the AAE and $f_{60}$ but not any of the sharp increases. Laboratory experiments and field observations have shown that the mass spectrum of OA from BB becomes increasingly like MO-OOA as it photochemically ages (Hennigan et al., 2011; Grieshop et al., 2009; Zhou et al., 2017). Therefore, aged BB aerosols can contribute to the MO-OOA factor (Bougiatioti et al., 2014). For BB2, all the BB tracers (AAE, $f_{60}$, BBOA, MO-OOA, CO, and eBC) are in good agreement. It seems that for these BB events, both BBOA and MO-OOA factors are needed to clearly describe the two plumes, while the HOA, CO and AAE facilitate the disentangling of the mixed combustion signal for BB1.

The eight AMS ion families at m/z < 120 (in different colors) and the elemental ratio of bulk OA (O/C, H/C, N/C and OM/OC) during BB1 and BB2 are included in Fig. S5. During BB1 and BB2, O/C, H/C and OM/OC were similar (O/C = 0.53 and 0.58, H/C= 1.37 and 1.34, and OM/OC= 1.84 and 1.90, respectively). Generally, O/C ratio ≥ 0.6 and H:C ratio ≥ 1.2 represent highly oxidized and highly saturated airmass (Brito et al., 2014; Brege et al., 2018; Tu et al., 2016; Zhou et al., 2017). Therefore, the observed elemental ratios during BB1 and BB2 in this study tend to agree well with the processed or oxidized airmass as reported in aged smoke plumes. Also, as evident in the MO-OOA factor, the consistency in elemental ratios between BB1 and BB2 shows that, regardless of the material burned, aerosols become chemically identical as they age and smoke plume gets more diluted (Jimenez et al., 2009; Ng et al.,

2010; Brito et al., 2014). Specific fragments can improve understanding of differences in OA composition and processing during BB1 and BB2, especially $f_{44}$ vs. $f_{43}$ (Fig. 5h) and $f_{44}$ vs. $f_{60}$ (Fig. 5i) (Ng et al., 2010; Cubison et al., 2011). The average $f_{44}$ values for BB1 (0.17) and BB2 (0.16) were similar, but BB2 had slightly higher $f_{43}$ than BB1 (0.059 vs. 0.051), which is consistent with BB2 being primarily composed of an aged BB plume. The two BB events generally overlapped on the $f_{44}$ vs. $f_{60}$ space. The progression of $f_{44}$ vs. $f_{43}$ and $f_{44}$ vs. $f_{60}$ as a function of time elapsed during BB2 are also shown in Fig. 5 (h-i). The observed direction of the trend during BB2 was similar to previous field studies, showing an increase in $f_{44}$ and a decrease in $f_{60}$ due to photochemical ageing (Cubison et al., 2011). Previous studies have shown that increase in $f_{44}$ with photochemical ageing may lead to the production of carboxylic acids (Zhang et al., 2005; Takegawa et al., 2007). There was not a discernible temporal progression in these relationships for BB1 (Figs. S5 i & j), which is likely attributed to the presence of mixed sources, including processed BB aerosols from different fire regions (as discussed in *Section 3.2*) and non-BB anthropogenic emissions (as discussed in *Section 3.1*).

Figure 5 (b-c & e-f) shows the average size distributions of NR-PM$_1$ species and key organic signals at m/z 43, 44, 55, 57 and 60 during BB1 and BB2. Organic signals m/z 43 and 44 were dominated by $C_2H_3O^+$, an ion fragment from oxidized organic compounds including aldehydes and ketones, and $CO_2^+$, an ion fragment from carboxylic acids, whereas m/z 55 and 57 were dominated by $C_4H_7^+$ and $C_4H_9^+$, respectively, which are ion fragments from hydrocarbons. The m/z 60 was primarily the AMS BB indicator, $C_2H_4O_2^+$, ion fragment of anhydrous sugar (e.g., levoglucosan). During BB2, the aerosol composition and the organic fragments showed a unimodal distribution, with a mode diameter in the accumulation mode size range of about 400 nm; the aerosol appears to be internally mixed. This confirms our previous discussion that the oxidized organic compounds ubiquitously dominated the aerosols during BB2, signifying the presence of aged BB emissions (Alfarra et al., 2004; Zhang et al., 2005; Chakraborty et al., 2015). During BB1, the size distribution of total nitrates, OA and organic mass fragments showed a similar distribution peaking at about 400 nm. However, the size distribution of sulfate and ammonium aerosols showed peaks at a significantly larger diameter of about 700 nm, and m/z 60, 55, and 57 showed enhanced signals at condensation mode (~100 - 200 nm) compared to those during BB2. Given the difference in the size distribution of aerosol composition during BB1, it appears to be externally mixed. The contribution of sulfate and ammonium to the NR-PM$_1$ composition for BB1 is also much greater than for BB2 (sum of 31 % and 19 %, respectively). The external mixing and the higher contribution of ammonium sulfate likely represents a mixture of organics from aged BB emission with an anthropogenic marine signal (e.g. inclusion of shipping activities and oil and gas extraction as discussed in earlier sections and in Zhou et al. (2023)). This marine signal in BB1 may be indicating that the Mexican fires transported over the Gulf of Mexico contributed to BB1 while the internal mixing and lack of marine signal in BB2 may indicate that the Central US fires dominated that period. However, the presence of mixed sources of processed BB aerosols and non-BB anthropogenic emissions at PA complicates the size distributions of NR-PM$_1$ composition for both BB1 and BB2 events, which needs further investigation.

The nitrate fraction of NR-PM$_1$ was similar during BB1 (2 %) and BB2 (3 %). Studies have confirmed that particulate organonitrates (ON) in the atmosphere are closely associated with BB emissions (Joo et al., 2019; Brege et al., 2018; Zhu et al., 2021; Tiitta et al., 2016). In this study, ON was observed and appeared to account for most of the $NO^+$ and

$NO_2^+$ (major ions of inorganic and organic nitrates in AMS) signals detected in NR-PM$_1$ during the BB periods. The signal ratios of $NO^+$ and $NO_2^+$ were 7.5 and 7.7 for BB1 and BB2, respectively, substantially higher than the ratio for pure ammonium nitrate particles ($R_{AN}$ = 2.38). Based on this information and following the method proposed by Farmer et al. (2010), we estimate that nearly all the $NO^+$ and $NO_2^+$ signals measured during the BB periods (~92% and ~99% for BB1 and BB2, respectively) were contributed by ON. The nitrate signal measured by the AMS that are attributed to ON, referred to as NO$_3$-ON, had similar trend as AAE and f$_{60}$ during the BB events (Fig. 2a). There was a clear enhancement in NO$_3$-ON concentration during BB1 and BB2 events (0.49 ± 0.20 and 0.79 ± 0.22 µg/m$^3$, respectively) compared to the marine background period (0.21 ± 0.13 µg/m$^3$). ON is a known chromophore and thus likely contributes to the increase in AAE during BB1 and BB2. Gas phase compounds like phenols produced during BB can undergo nitrate-mediated oxidation to form aqueous phase SOA (Xiao et al., 2022). Laboratory studies have demonstrated that aqueous phase SOA from BB are chromophores and can influence the aerosol light absorption properties (Pang et al., 2019; Jiang et al., 2021; Smith et al., 2014). Additionally, BB emission can also undergo rapid oxidation by nitric radicals during nighttime to form SOA (Lalchandani et al., 2022). If we assume that the UV absorption from ON was solely responsible for enhancement in AAE during BB events, we can evaluate the relationship between AAE and ON during BB1 and BB2 (Fig. 5g). We do see a high correlation between these two during BB1 (r$^2$= 0.85) and BB2 (r$^2$= 0.95), when the correlation line is forced through zero. Thus, we observe potential indication of BrC as represented by ON during the BB events. This assumption may be an over-simplification, as other BrC compounds may also contribute to the UV absorption in these plumes.

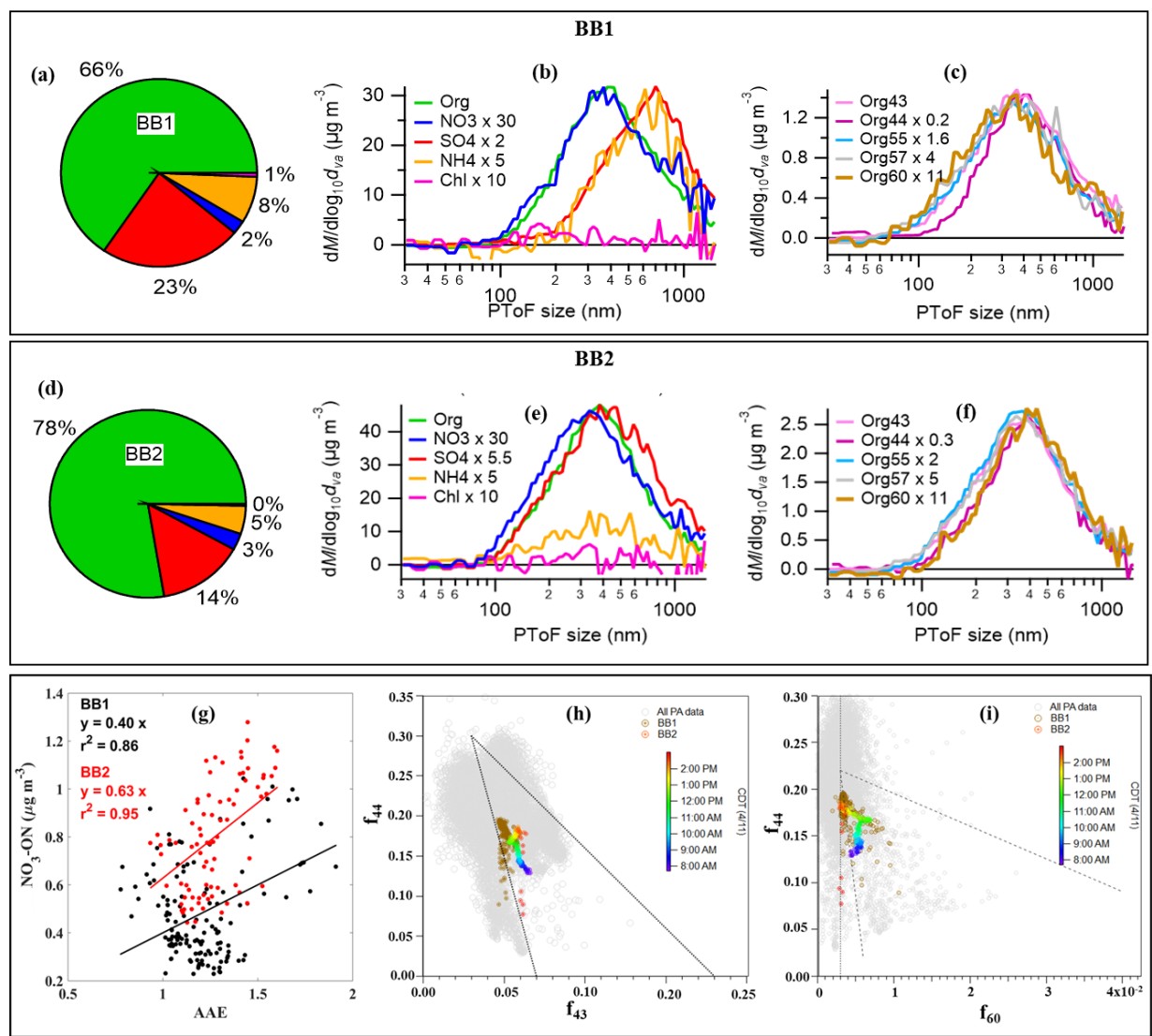


**Figure 5. (a)** Pie chart showing NR-PM$_1$ composition during BB1; **(b-c)** average particle time-of-flight (PToF) size distribution of NR-PM$_1$ species and select organic m/zs during BB1; **(d)** pie chart showing NR-PM$_1$ composition during BB2; **(e-f)** average size distribution of NR-PM$_1$ species and select organic m/zs during BB2; **(g)** scatter plot of NO$_3$-ON vs AAE during BB1 and BB2; scatterplot of **(h)** f$_{44}$ vs. f$_{43}$ and **(i)** f$_{44}$ vs. f$_{60}$, where BB2 data are colored as a

function of time of the day and the grey markers correspond to the measured OA during this study. The triangular boundaries in **panel h** and **panel i** represent the ranges set by Cubison et al, (2011) and Ng et al. (2010). The vertical dashed line in **panel i** denote f$_{60}$ = 0.003. In the pie-charts (**panels a** and **d**): organics (in green), nitrates (in blue), sulfates (in red), ammonium (in yellow) and chloride (in magenta).

### 3.4. Application of acetonitrile as VOC tracer for biomass burning

BB emissions consist of a mixture of organic and inorganic compounds in the gas and aerosol phase (Holzinger et al., 1999). As with the aerosol fraction, there are common VOC markers for BB; acetonitrile has been utilized as a BB tracer in PTR-MS measurements (de Gouw et al., 2003a; Karl et al., 2003; Sinha et al., 2014). BB emissions are a significant component of the global budget of acetonitrile (de Gouw et al., 2003a; Holzinger et al., 1999, 2001). However, studies have highlighted that acetonitrile signals from BB can be convoluted by local vehicular emissions

(Guan et al., 2020; Swarthout et al., 2013), coal-burning (Jobson et al., 2010; Valach et al., 2014; Inomata et al., 2013) and industrial emissions including oil and gas activities (Cai et al., 2019). In fact, background acetonitrile levels have been reported to vary from ~100 pptv to above 600 pptv across different regions (Huangfu et al., 2021 and references therein). Therefore, using acetonitrile in an anthropogenically-influenced environment like PA requires careful consideration (Huangfu et al., 2021). In this section we evaluate the efficacy of acetonitrile as a BB tracer for dilute

plumes on the Gulf Coast of Texas.

     During the two BB events identified in this study, the acetonitrile did not follow the same trend as AAE or $f_{60}$ (Fig. 6a & Fig. 2). Acetonitrile had peaks prior to the onset of BB1 and in the early afternoon of April 10 with relatively low concentrations during BB2. Thus, we hypothesize that the acetonitrile levels at the study site are impacted by emissions from the dense network of on-shore and off-shore oil and gas activities in the PA/Corpus Christi region and/or urban

background during continental airmass regimes (Fig. 1c). To determine whether the acetonitrile mixing ratios associated with the observed BB plumes at PA would exceed the local background, we (i) investigated the geospatial variability of acetonitrile emissions (including industrial and traffic sources) in Corpus Christi by evaluating mobile measurements and (ii) estimated the acetonitrile in the BB events using an enhancement ratio of CO with respect to acetonitrile from the literature (Warneke et al., 2006).

The mobile measurement shows that while the ambient acetonitrile concentration varied on a daily basis, acetonitrile concentration was clearly enhanced in the major industrial sector of Corpus Christi (Fig. 6). The acetonitrile concentration in PA did not surpass the average measured acetonitrile from the major industrial sector of Corpus Christi in either BB event (Fig. 6a). These results indicate that local anthropogenic emissions likely enhance the background acetonitrile level in PA. Additional investigation is needed to characterize and define these local sources.

To further test acetonitrile as a BB tracer for this study, we estimated the BB-associated acetonitrile using an orthogonal regression-based equation formulation by Warneke et al. (2006). Because we observed CO enhancement during BB1 and BB2, we reorganized the equation to estimate acetonitrile concentration during the BB events (Eq. (3) below) using the literature-based $ER_{CO\text{-}acetonitrile}$ and the ambient CO measurements from our investigation. We acknowledge the limitation of this calculation which assumes: (i) the entire CO enhancement above background is

from the BB influence and (ii) $0.36 \pm 0.06$ ppbv of CO per pptv of acetonitrile is observed in the BB plume in an urban environment. Based on the standard deviation of the reported $ER_{CO\text{-}acetonitrile}$, we estimate that this calculation for BB related acetonitrile is ~17%.

$$Estimated\ Acetonitrile = \left(\frac{Ambient\ CO - CO\ background}{ER_{CO-acetonitrile}}\right) - Acetonitrile\ background \qquad (3)$$

where CO background = 75 ppbv, Enhancement Ratio ($ER_{CO\text{-}acetonitrile}$) = 0.36 ppbv and acetonitrile background = 0.115

ppbv were used as reported in Warneke et al. (2006).

Fig. 6a shows a timeseries of ambient acetonitrile measured during the campaign and the estimated acetonitrile concentration using the above-mentioned Eq. (3). The observed and estimated acetonitrile were similar during BB2, indicating that the observed acetonitrile was potentially influenced by the BB2 plume. However, the estimated acetonitrile did not match the observed acetonitrile during BB1 or during the preceding period of continental influence.

The observed acetonitrile was higher than estimated acetonitrile during the preceding period of continental influence, likely indicating the local industrial sources of acetonitrile as mentioned above. However, in the evening of April 10, the calculated acetonitrile was well above the ambient acetonitrile levels. This switch to condition of estimated acetonitrile greater than observed acetonitrile likely indicates a local combustion source that emits CO but does not emit acetonitrile and is therefore likely not BB. This analysis highlights that both CO and acetonitrile can be impacted

by local sources and specifically that acetonitrile cannot be used as a unique BB tracer for dilute BB plumes when the background acetonitrile level is high due to the presence of local anthropogenic sources. Overall, this study demonstrates that AAE and aerosol composition served as reliable indicators of transported BB plumes in the urban environment.

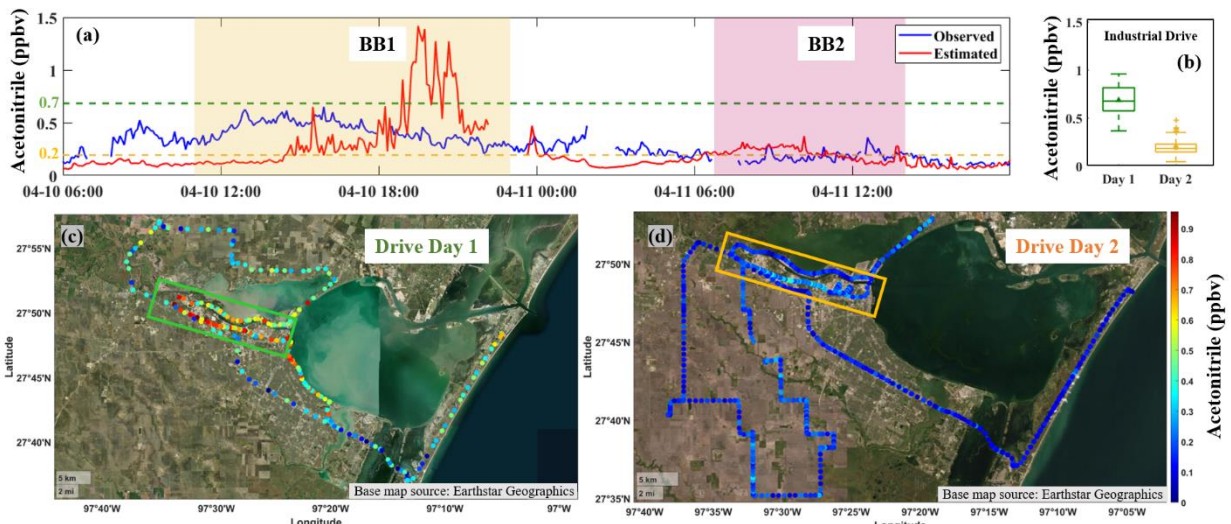

**Figure 6. (a)** Time series for ambient acetonitrile observed during the campaign (in blue) and estimated acetonitrile (in red) using the reorganized equation from Warneke et al. (2006); **(b)** box plot showing the acetonitrile concentration measured in the industrial corridor during mobile measurements on drive day 1 (April 16) and drive day 2 (April 18); and **(c-d)** acetonitrile concentration measured during mobile measurements on April 16 and 18 in Corpus Christi. The location of major industrial corridor is marked by the green box (**panel c**) and orange box (**panel d**). The dashed lines

in green and orange (**panel a**) represent average acetonitrile concentration measured in the major industrial corridor during drive days 1 and 2, respectively. The spatial distribution of acetonitrile (**panel c** and **d**) was created in MATLAB.

### 3.5. Evaluation of additional VOCs and trace gases during biomass burning events

Figure 7 shows the time series of acetonitrile, acetaldehyde, benzene, toluene, formaldehyde (HCHO), $O_3$, NO, $NO_2$, and $NO_x$. Notably, during BB events, benzene and toluene concentrations exhibited significant enhancement above the marine background levels, with increases of 0.23 ppbv and 0.19 ppbv (benzene), and 0.25 ppbv and 0.33 ppbv (toluene) observed for BB1 and BB2, respectively. Furthermore, elevated levels of $NO_x$ were observed during BB events compared to the marine background conditions, with enhancements of 2.81 ppbv and 2.49 ppbv noted for BB1 and BB2, respectively. Although the concentrations of these compounds were also slightly elevated above continental airmass periods, the magnitudes were comparatively lower.

Regarding acetaldehyde and HCHO, their concentrations surpassed the marine background levels; however, their profiles exhibited a photochemical production trend that correlated with the typical behavior observed for $O_3$. It should be noted that the measurement period was characterized by high ambient temperatures, which adds complexity to attributing the elevations of HCHO and acetaldehyde solely to BB plumes but does indicate enhanced photochemical activity within the BB plume. Characterizing VOCs and $NO_x$ during dilute BB plumes in an industrialized location is complicated due to the presence of multiple local emission sources and atmospheric processing during the transport. However, the identification of the BB plumes with AAE, AMS-driven PMF factors, $f_{60}$, and satellite imagery enables attribution of these additional pollutants to the same BB plume.

In addition to acetonitrile, other VOCs like phenols, furans, furfurals, and hydrogen cyanide have also been used as BB tracers (Tripathi et al., 2022; Coggon et al., 2016; Bruns et al., 2017). However, these VOCs were not included in the select list of measured compounds. It is known that furans, furfurals, and phenols are emitted in higher quantities by biomass burning compared to vehicular emissions, making them potentially important tracers for BB plumes (Mohr et al., 2013; Coggon et al., 2019; Yuan et al., 2017; Wang et al., 2020). However, these compounds exhibit high atmospheric reactivity and undergo secondary transformations. Previous studies have demonstrated the presence of secondary oxidation products of these compounds such as maleic anhydride and nitrophenols in BB plumes (Yuan et al., 2017; Wang et al., 2020). Further, Lalchandani et al. (2022) reported a regional impact of furans from BB, leading to increased levels of SOA precursors such as ammonium nitrate and BBOA. SOA evolutions are remarkably higher during the smoldering than in flaming phase, due to the higher emission of VOCs (Li et al., 2021). Considering the high atmospheric reactivity and potential for SOA formation of these primary VOCs, further investigations into their secondary oxidized products and associated SOA are warranted in aged BB plumes. In the current study, the enhancement of MO-OOA, HCHO and acetaldehyde indicate that the plume had undergone photochemical oxidation during transport which likely would have made the measurement of reactive VOC tracers difficult.

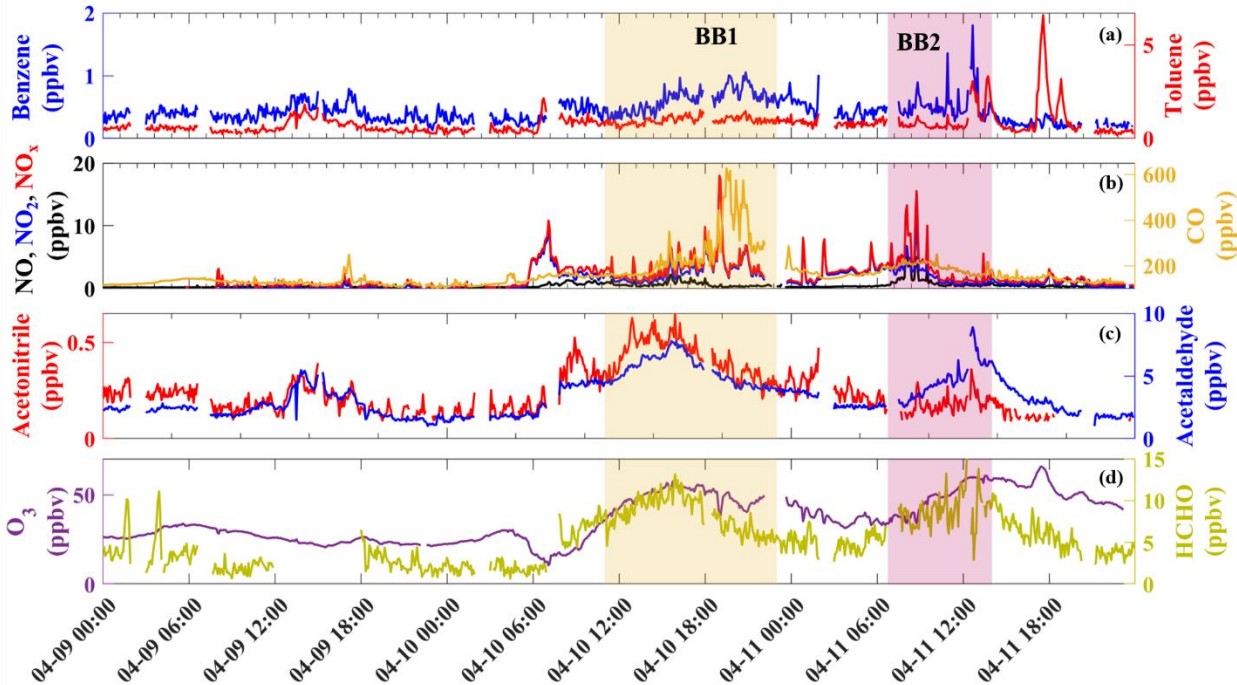

**Figure 7.** Time series of **(a)** benzene and toluene; **(b)** NO, NO$_2$, NO$_x$ and CO; **(c)** acetonitrile and acetaldehyde; **(d)** O$_3$ and HCHO.

## 4. Atmospheric Implication and Outlook

Previous studies have demonstrated that transported BB plumes advected to the surface can contribute to O$_3$ production and lead to exceedances of the National Ambient Air Quality Standard (NAAQS) (Wilkins et al., 2020; Schade et al., 2011; McMillan et al., 2010; Lei et al., 2018; Langford et al., 2015). For instance, transported BB emission was estimated to contribute ~10 ppbv to two O$_3$ exceedance days in Las Vegas during summer of 2013 (Langford et al., 2015). Similarly, Wilkins et al. (2020) asserted that aged BB plumes were more O$_3$ enriched and reported that aged plumes (4 - 7 days) contributed, on average, 15 ppbv to surface O$_3$ in the Midwestern US. Lei et al. (2018), McMillan et al. (2010) and Schade et al. (2011) showed the transport of CO and O$_3$ from fires in the US Pacific Northwest to Houston, which then contributed to an O$_3$ exceedance period in the Houston area. These results highlight that BB contribution can be an important factor in urban O$_3$ chemistry.

During the BB events discussed in this study, the Texas Commission on Environmental Quality (TCEQ) O$_3$ monitors recorded elevated O$_3$ concentrations across Southern Texas. Figure 8a shows the time series of O$_3$ concentration for April 9 - 11 in Southern Texas including Houston-Galveston, San Antonio, and Corpus Christi. Three representative sites from each of these metropolitan areas are plotted to highlight concurrent enhancement in O$_3$ on those days. There were more than thirty other TCEQ sites that showed elevated O$_3$ concentration above 65 ppbv during this time period (https://www.tceq.texas.gov/gis/geotam-viewer). The presence of smoke in Houston can be evaluated using the Black and Brown Carbon (BC)$^2$ network. (BC)$^2$ is a TCEQ-funded aerosol optical network in Texas operated by our research group that utilizes the same aerosol optical measurement instrumentation as in this study. Elevated AAE was also

observed on April 10 - 12 at $(BC)^2$ in Houston (Fig. 8b). These results indicated that the smoke distribution from the
NOAA HMS product (Fig. 4) may have had regional impacts on $O_3$ concentrations across Southern Texas. However, accurately estimating the contribution from BB emission to the local $O_3$ enhancement with single-point measurements is difficult (Thompson et al., 2019). Additional investigation of this regional event is needed to confirm the potential BB contribution to urban air quality.

Permanent ground-based air quality monitoring networks play a crucial role in identifying such events but often lack specificity towards identification of BB influence. In this regard, the low-cost aerosol optical measurements in this study exhibited exceptional ability to identify BB events even in a dilute plume in an industrialized urban environment. The results of this study support the implementation of an extended network of low-cost aerosol optical measurements to identify the influence of BB plumes, especially in cities that are designated as non-attainment or marginal nonattainment of criteria air pollutants.

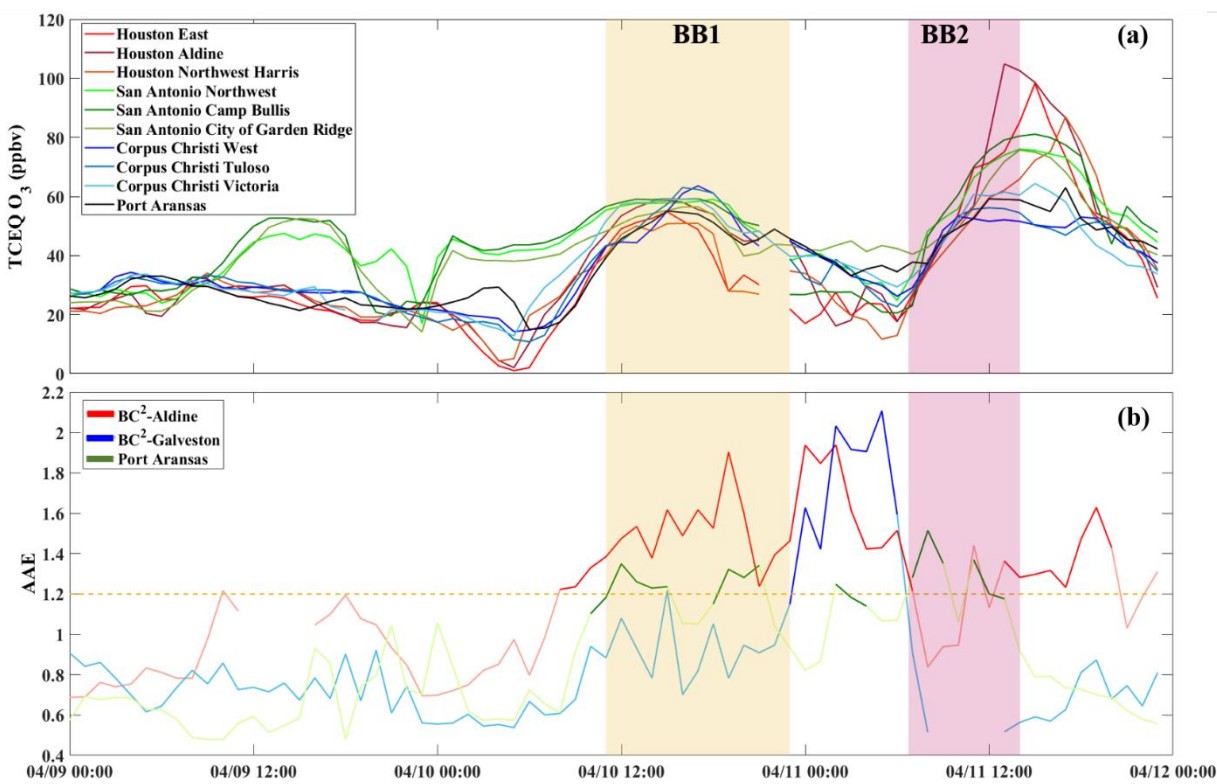

**Figure. 8.** Time series of **(a)** $O_3$ concentrations reported by TCEQ sites and PA (this study); **(b)** AAE observed at $(BC)^2$-sites in Houston and PA. AAE above the threshold for BB indication (campaign average + 2 standard deviation) is plotted in darker color. Hourly data from the different air quality monitors including $O_3$ from the TCEQ monitors are available from the TCEQ website (https://www.tceq.texas.gov/gis/geotam-viewer).

**Data availability**

The data used in this study can be accessed through the publicly available link: https://doi.org/10.18738/T8/DEPG3R.

## Author Contributions

SS, MM, MG, SY, SA, SZ, FG, CYC, JHF, RJS, SU and RJS participated in the field campaign, including measurements and data quality assurance. S.S. performed data analysis. R.J.S. supervised the project and data analysis. S.S. prepared a draft of the manuscript. S.S., and R.J.S. edited the final version of the manuscript. All authors reviewed the manuscript and provided inputs for data analysis.

## Competing Interest

The authors declare that they have no conflict of interest.

## Acknowledgements

We would like to thank Dr. J. David Felix of Texas A&M Corpus Christi for assistance with the filter sampling at TAMU-CC and Johnathan White of Baylor University for assistance with the OCEC analysis. The preparation of this manuscript was financed through a grant from the Texas Commission on Environmental Quality (TCEQ, Grant number: 582-18-81339), administered by the University of Texas at Austin, Center for Energy and Environmental Resources (CEER) through the Air Quality Research Program (AQRP). The black and brown carbon (BC)[2] monitoring in Houston and El Paso in 2021 Ozone season was financed through the TCEQ (Grant number: 582-18-81339). The content, findings, opinions, and conclusions are the work of the authors and do not necessarily represent findings, opinions, or conclusions of the TCEQ or the AQRP. The authors gratefully acknowledge the NOAA Air Resources Laboratory (ARL) for the provision of the HYSPLIT transport and dispersion model and/or READY website (http://www.ready.noaa.gov) used in this publication. The authors acknowledge FracTracer Alliance for the permission to use GIS map of oil and gas activities in Texas.

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

**Supplementary Material for**

**Evaluation of aerosol- and gas-phase tracers for identification of transported biomass burning emissions in an industrially influenced location in Texas, USA**

5  Sujan Shrestha[1], Shan Zhou[2,3], Manisha Mehra[1], Meghan Guagenti[1], Subin Yoon[2], Sergio L. Alvarez[2], Fangzhou Guo[2,3,a], Chun-Ying Chao[3], James H. Flynn III[2], Yuxuan Wang[2], Robert J. Griffin[3,4], Sascha Usenko[1], Rebecca J. Sheesley[1]

[1]Department of Environmental Science, Baylor University, Waco, TX, USA
[2]Department of Earth and Atmospheric Sciences, University of Houston, Houston, TX, USA
[3]Depertment of Civil and Environmental Engineering, Rice University, TX, USA
10  [4]School of Engineering, Computing, and Construction Management, Roger Williams University, Bristol, RI, USA
[a]now at Aerodyne Research Inc., Billerica, WA, USA
Correspondence to: Rebecca J. Sheesley (Rebecca_Sheesley@baylor.edu)

### 1. Equivalent BC (eBC) calculation

Filter-based optical techniques of black carbon (BC) measurement do not measure the mass concentration directly but uses Mie theory to measure the light absorption coefficient of particles. The absorption coefficients ($\sigma_{abs}$) are converted into an equivalent mass concentration (eBC) using mass absorption cross section (MAC) (equation (1) below). Field based studies have shown a large variability in MAC values ranging from 1.6 $m^2g^{-1}$ to 28.3 $m^2g^{-1}$ at 550 nm (Sharma et al., 2002; Bond and Bergstrom, 2006). Large temporal and spatial variability in MAC value reported in previous studies is due to different mixing states of BC. Once emitted into the atmosphere these particles are subject to several coating processes with layers of other organic and inorganic materials (Zhao et al., 2021; Bond and Bergstrom, 2006). The coating over a core BC enhances the aerosol absorption by acting as a lens that helps in focusing more incident light on the enclosed BC core (a phenomena known as "lensing effect") (Fuller et al., 1999). This results into higher MAC value of the coated BC. Bond and Bergstrom (2006) reported MAC value of 7.5 ± 1.2 $m^2g^{-1}$ at 550 nm for a fresh uncoated particle. But due to coating over fresh BC during local and long-range transport, the absorption can be enhanced by up to 100% (Schwarz et al., 2008; Bond and Bergstrom, 2006). Some of the commercially available instruments, however, use a fixed MAC values to obtain BC concentration, e.g., an aethalometer uses 7.77 $m^2g^{-1}$ at 880 nm or 13.14 $m^2g^{-1}$ at 520 nm (Drinovec et al., 2015). Use of fixed MAC values is convenient when additional collocated measurements required for MAC calculation are not available but this approach remains debatable (Zhao et al., 2021)

$$\text{eBC } (\mu g\ m^{-3}) = \frac{\sigma_{abs}\ (Mm^{-1})}{MAC\ (m^2g^{-1})} \qquad [1]$$

During the field measurement in Port Aransas, $\sigma_{abs}$ was measured using Tri-color Absorption Photometer (TAP) through a $PM_{2.5}$ cyclone at three wavelengths: 365 nm, 520 nm and 640nm. Operational detail about TAP are presented elsewhere (Bernardoni et al., 2021; Ogren et al., 2017). In addition to aerosol optical measurement, a $PM_{2.5}$ bulk filter sampler was operated at Texas A&M Corpus Christi campus which is ~16 miles areal distance apart from the Port Aransas site. During the campaign, a total of six $PM_{2.5}$ bulk samples were collected. Details about the filter sample are presented in Table S1. The organic and elemental carbon (OC and EC) in the filter samples were measured using Sunset OC EC analyzer using NIOSH protocol (Schauer, 2003). Both Port Aransas and Texas A&M Corpus Christi sites have close proximity to the Gulf of Mexico and received airmasses predominantly from the East and Southeast direction during the campaign. Therefore, it is realistically more appropriate to use the MAC derived as a slope of the linear regression between $\sigma_{abs}$ from TAP and EC concentration from the $PM_{2.5}$ filter samples collected during the campaign compared to using literature values. Using this method, the derived MAC at 520 nm was 11.45 ± 5.32 $m^2g^{-1}$ which is slightly lower than that used by aethalometer (13.14 $m^2g^{-1}$). Mathematically, the MAC is given by the following equation:

$$MAC = \frac{\sigma_{abs}\ (Mm^{-1})}{EC\ (\mu gm^{-3})} \qquad [2]$$

where $\sigma_{abs}$ is the average absorption coefficient measured by TAP at 520 nm during the $PM_{2.5}$ filter sampling period.

## 2. Positive Matrix Factorization of Organic Aerosol Matrix and Combined Organic and Inorganic Aerosol Matrices

To investigate the sources and processes of organic aerosols (OA), we performed positive matrix factorization (PMF) analysis on the high-resolution mass spectra (HRMS) of 1) organics only and 2) the combined spectral matrices of organic and inorganic species, respectively using the PMF2 algorithm in robust mode (Paatero and Tapper, 1994). We first generated the ion-speciated HRMS matrix and the corresponding error matrix from PIKA, and then analyzed using the PMF Evaluation Tool v3.06B (Ulbrich et al., 2009). We did PMF analysis on the entire sampling period

covering both the stationary measurements and the mobile measurements.

For the organic PMF analysis, the OA data and error matrices were refined prior to PMF analysis according to the protocol summarized previously (Zhang et al., 2011; Ulbrich et al., 2009). Ions with m/z up to 190 were included in the PMF analysis. Isotopes were removed to avoid giving excess weight to their parent ions. Noisy ions were removed from the data matrix. These treatments largely improved the OA factorization but had negligible impact on the mass

concentrations. A minimum error was introduced for each ion. The "bad" ions with S/N ratio < 0.2 were downweighed by increasing their error values by a factor of 10, while the "weak" ions with S/N between 0.2 and 2 were downweighed a factor of 2 as described by Ulbrich et al. (2009). $O^+$, $OH^+$, $H_2O^+$, and $CO^+$ ions were also down weighted to avoid additional weight to $CO_2^+$, as their signals were all scaled to that of $CO_2^+$. PMF solutions were tested from 2 to 7 factors, and the rotational forcing parameter, fPeak, varied between -1 and 1 (step = 0.2).

We also performed PMF analysis on the combined spectral matrices of organic and inorganic species of the HR-AMS (Zhou et al., 2017; Zhang et al., 2011; Paatero and Tapper, 1994). PMF is commonly applied to the organic mass spectral matrix to determine distinct OA factors (Zhang et al., 2011). However, conducting PMF analysis on the combined spectra of organic and inorganic aerosols allows for the derivation of additional information. In this study, we performed PMF analysis on the combined HR spectral matrices of organic and inorganic species. Organic ions at

m/z 12 – 180 and major inorganic ions, i.e., $SO^+$, $SO2^+$, $HSO_2^+$, $SO_3^+$, $HSO_3^+$, and $H_2SO_4^+$ for sulfate; $NO^+$ and $NO_2^+$ for nitrate; $NH^+$, $NH_2^+$, and $NH_3^+$ for ammonium; and $Cl^+$ and $HCl^+$ for chloride were included, and the ion signals were expressed in nitrate-equivalent concentrations. The error matrix was pretreated the same as the PMF analysis of organic matrix only. After PMF analysis, the mass concentration of each OA factor was derived from the sum of organic signals in the corresponding mass spectrum after applying the default RIE for organics (1.4) and the time

dependent CDCE. The solutions for two to nine factors were explored at a fixed rotational parameter (FPEAK = 0). We performed similar evaluation procedures as to the organic PMF analysis and chose the seven-factor solution as the optimum solution for the combined PMF analysis. Following the procedures listed in Table 1 of (Zhang et al., 2011), all PMF solutions have been evaluated by investigating the key diagnostic plots, mass spectra, correlations with external tracers and diurnal profiles. We selected the seven-factor solution with fPeak = 0 from the PMF analysis

of the combined matrices as the optimum solution. The solution is presented and discussed in detail below.

After a detailed evaluation of temporal trends, mass spectral profiles, and correlations with ions, we identified seven distinct OA factors. These seven factors are: 1) hydrocarbon-like organic aerosol (HOA) that is associated with traffic related primary emission, 2) biomass burning organic aerosol (BBOA) associated with campfires as well as regional transported wildfire plumes, 3) less-oxidized oxygenated organic aerosol (LO-OOA) representing less processed and

fresher secondary organic aerosol (SOA) (O/C = 0.51), 4) more-oxidized OOA (MO-OOA) possibly representing more processed and aged SOA (O/C = 1.22), 5) an OOA that was associated with ammonium nitrate and biomass burning (AN-BB-OOA), 6) a highly oxidized OOA associated with ammonium sulfate (AS-OOA), and 7) a highly oxidized OOA associated with acidic sulfate (acidic-OOA). Three of these factors had inorganic signals in the mass spectra, and were associated with neutralized ammonium nitrate, neutralized ammonium sulfate, and acidic sulfate

signals, respectively.

**Table S1.** Details about PM$_{2.5}$ filter samples collected at Texas A&M Corpus Christi

| Sample ID | Start Date | Start Time | End Date | End Time |
|---|---|---|---|---|
| 210403_MV2.5_COR | 4/3/21 | 10:06 | 4/6/21 | 17:02 |
| 210406_MV2.5_COR | 4/6/21 | 19:19 | 4/9/21 | 18:43 |
| 210409_MV2.5_COR | 4/9/21 | 18:59 | 4/13/21 | 11:41 |
| 210413_MV2.5_COR | 4/13/21 | 11:48 | 4/16/21 | 18:21 |
| 210416_MV2.5_COR | 4/16/21 | 18:38 | 4/20/21 | 17:01 |
| 210420_MV2.5_COR | 4/20/21 | 17:07 | 4/22/21 | 11:34 |

**Table S2.** The 1-min and 2.5-min minimum detection limits (MDL) of the measured nonrefractory submicron aerosol species during the sampling campaign, which were determined as three times the standard deviation (3σ) of the corresponding signals in particle-free ambient air.

| Species | MDL- 2.5 min ($\mu g\ m^{-3}$) |
|---|---|
| Organics | 0.37 |
| Sulfate | 0.072 |
| Nitrate | 0.02 |
| Ammonium | 0.063 |
| Chloride | 0.024 |

**Table S3.** Minimum detection limit (MDL) for 30-s averaged data and associated uncertainty for trace gas measurements.

| Species | Uncertainty (%) | MDL (ppbv) |
|---|---|---|
| CO | 1.4 | 0.13 |
| $CO_2$ | 1 | 0.33 |
| NO | 4.7 | 0.07 |
| NO | 4.5 | 0.35 |
| $NO_2$ | 5.4 | 0.13 |
| $NO_2$ | 9.3 | 0.49 |
| $NO_y$ | 5.6 | 0.48 |
| $O_3$ | 1.9 | 0.23 |
| $SO_2$ | 9.1 | 0.92 |

**Table S4.** Minimum detection limit (MDL) in ppbv and uncertainty associated with the measured VOCs during the sampling campaign.

| Species | m/z | Uncertainty (%) | MDL (ppbv) |
|---------|-----|-----------------|------------|
| Formaldehyde | 31 | 10.8 | 0.66 |
| Acetonitrile | 42 | 10.7 | 0.09 |
| Acetaldehyde | 45 | 9.6 | 0.26 |
| Acetone | 59 | 20.9 | 0.42 |
| DMS | 63 | 9.6 | 0.15 |
| Isoprene | 69 | 10.1 | 0.15 |
| MVK+MACR | 71 | 9.5 | 0.16 |
| MEK | 73 | 9.6 | 0.12 |
| Benzene | 79 | 9.9 | 0.13 |
| Toluene | 93 | 9.9 | 0.16 |
| Monoterpene | 137 | 11.2 | 0.52 |
| Hydroxyacetone | 75 | 16.7 | 0.44 |
| Styrene | 105 | 11.4 | 0.1 |
| Xylene | 107 | 11.1 | 0.18 |

**Figures**

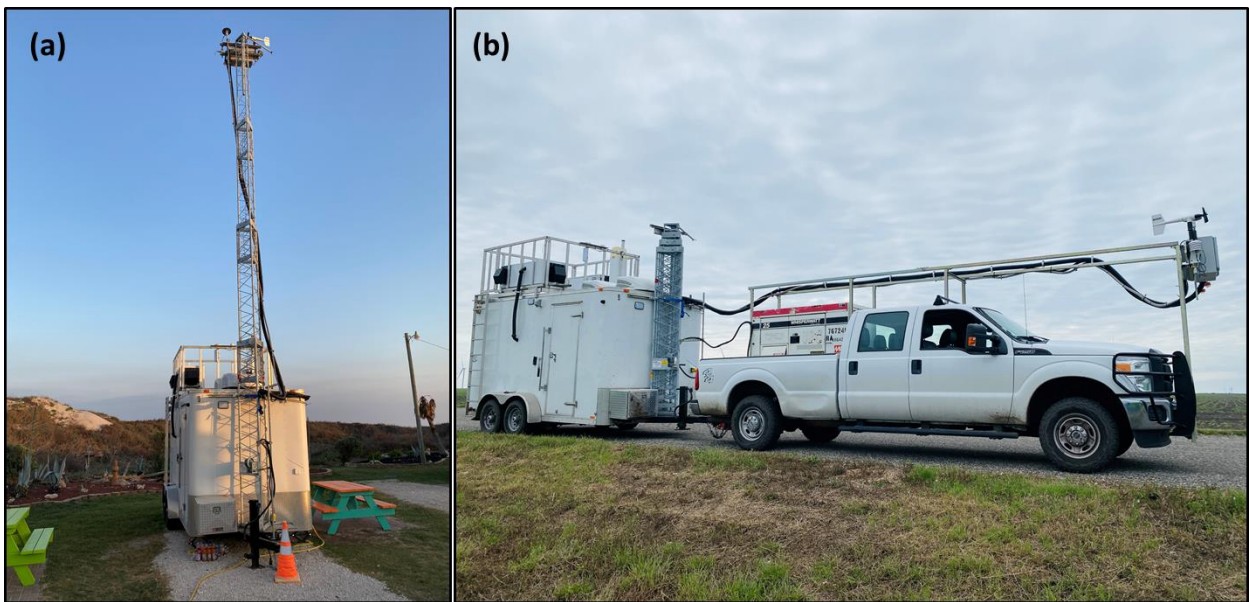

**Fig. S1.** Mobile air quality laboratory (MAQL2) during **(a)** stationary phase and **(b)** mobile phase.

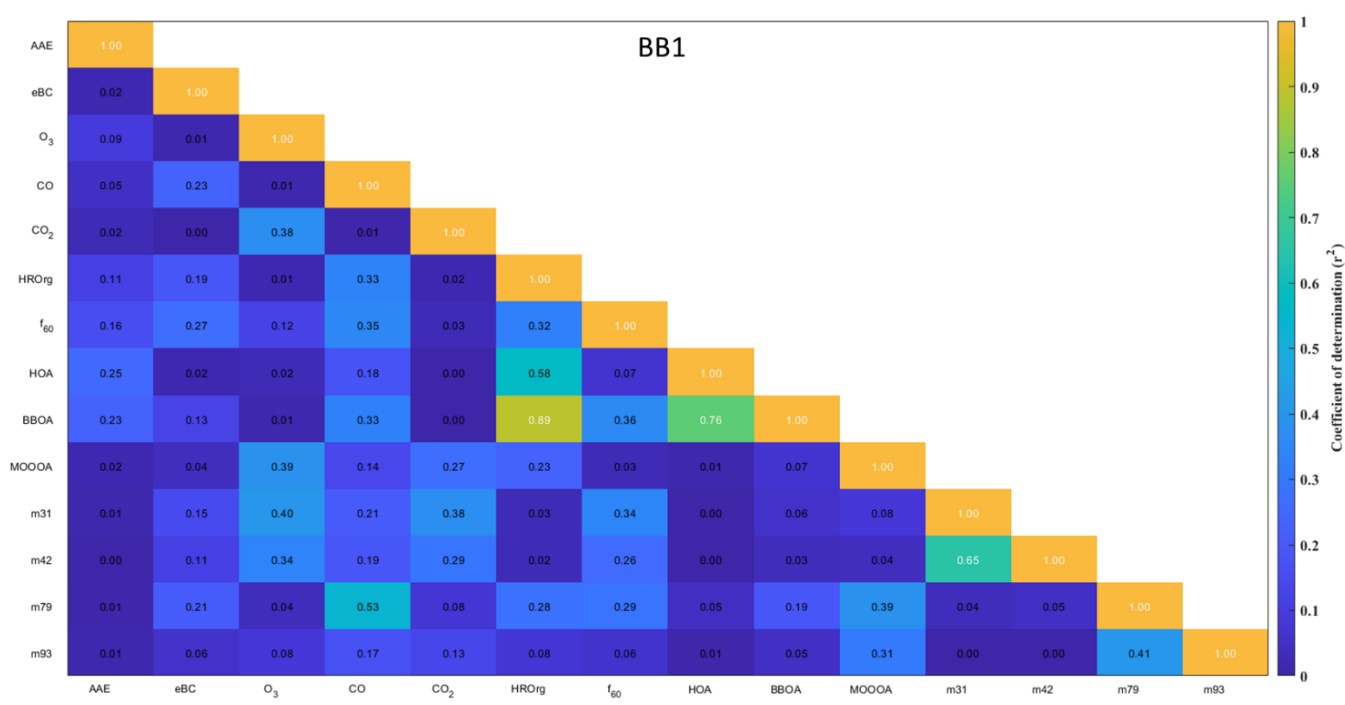

**Fig. S2.** Correlation plot of select-aerosol optical properties, trace gases, aerosol composition and VOCs during BB1.


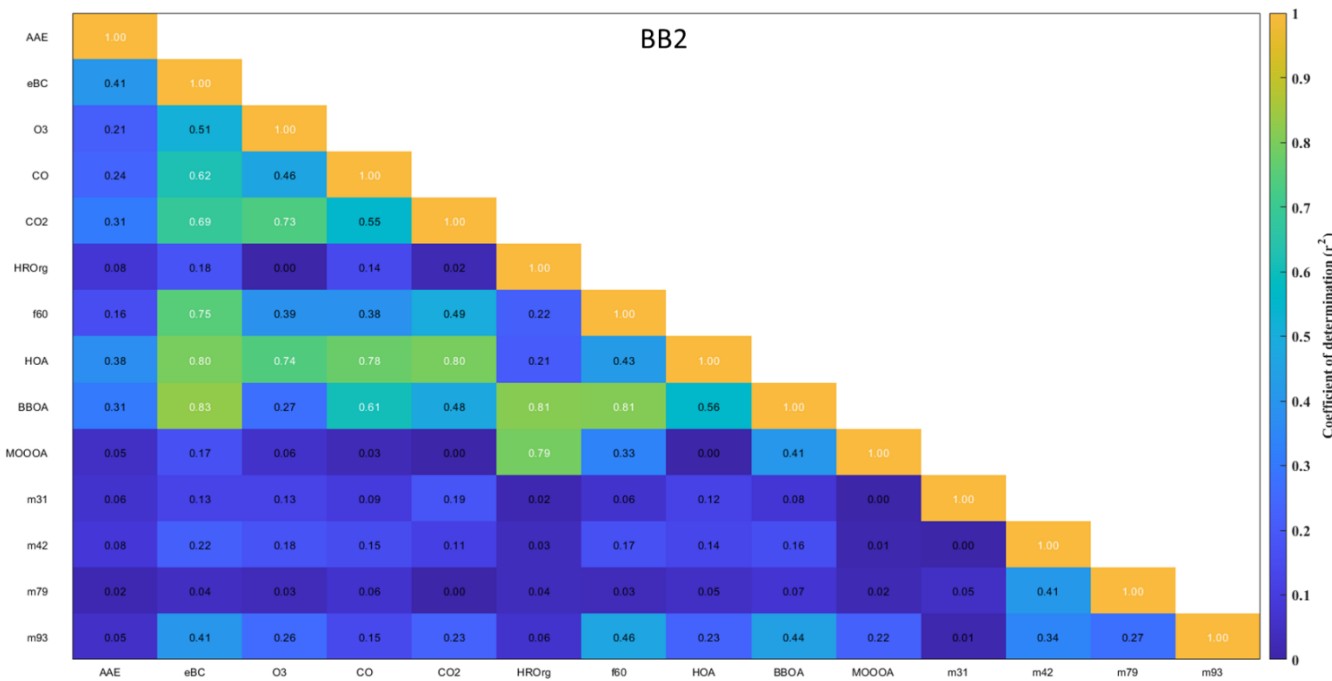

**Fig. S3.** Correlation plot of select-aerosol optical properties, trace gases, aerosol composition and VOCs during BB2.

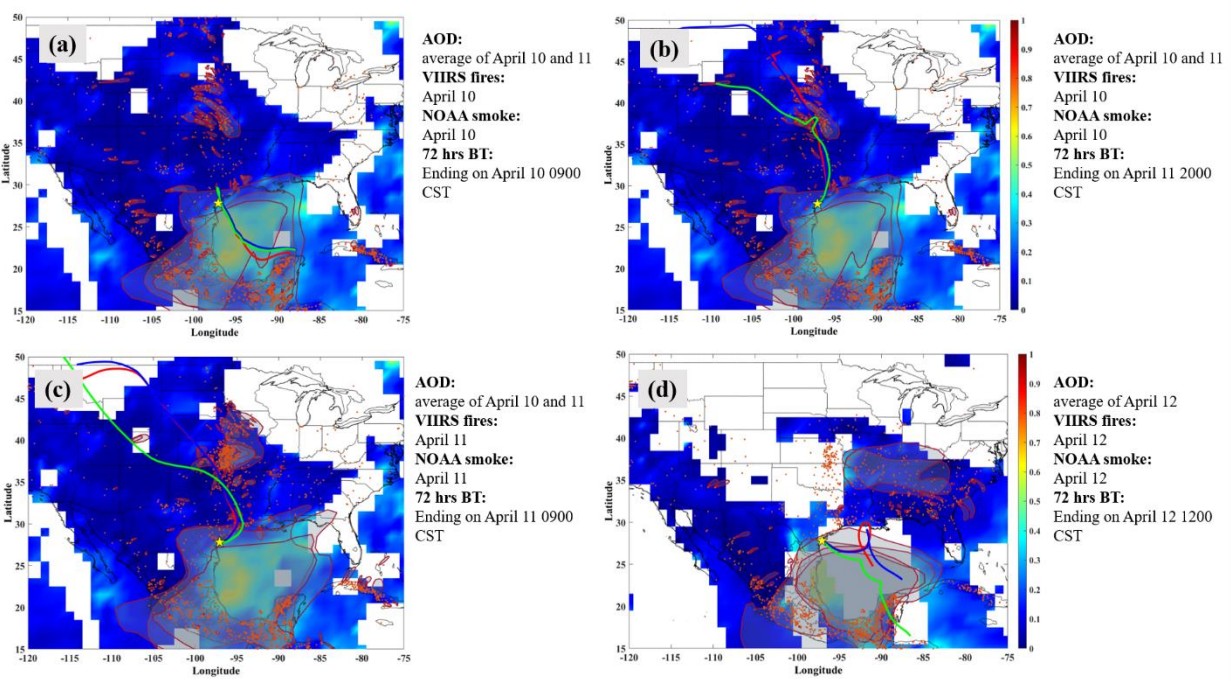

**Fig. S4.** Spatial distribution of average Aerosol Optical Depth (AOD) from Aqua and Terra satellites (April 10 – 12, 2021). Visible Infrared Imaging Radiometer Suite (VIIRS) active fire, NOAA Hazard Mapping System (HMS) smoke and HYSPLIT Backward trajectories (BTs) at different starting heights: 50 m (red), 100 m (blue) and 500 m (green)
are included in the map. The ending times of the BTs are chosen to show the gradual change in the path of BTs from the Central Mexico to the Northern US during the period of interest in this study. The study site Port Aransas is denoted by a yellow star symbol.

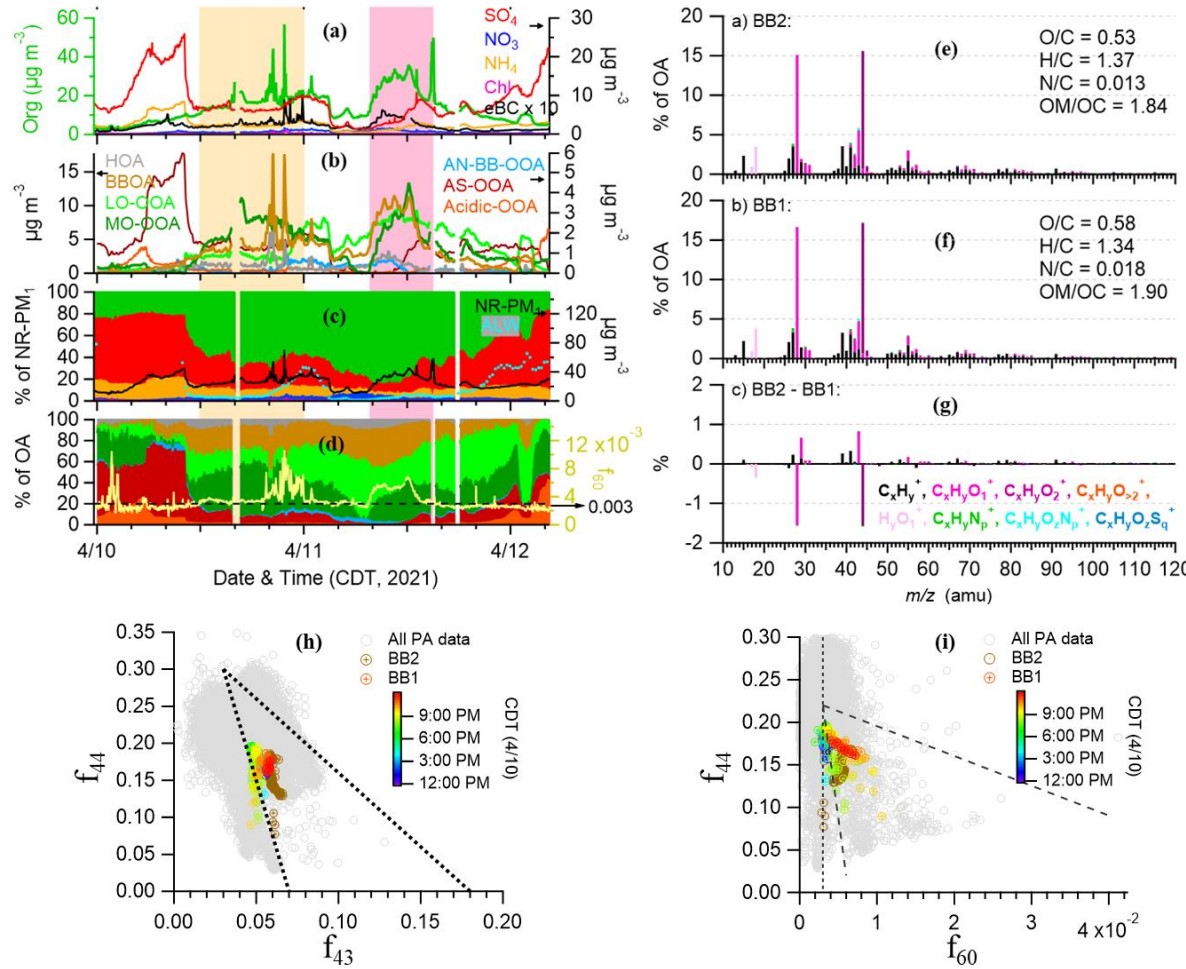

**Fig. S5.** Time series of **(a)** mass concentrations of NR-PM$_1$ species, **(b)** mass concentrations of OA factors determined from PMF analysis, **(c)** NR-PM$_1$ composition, **(d)** OA composition and f$_{60}$ (i.e., C$_2$H$_4$O$_2^+$ / OA) is the yellow lines, and; **(e-g)** high resolution mass spectra (HRMS) of OA during two BB periods and the difference HRMS colored by eight ion families at m/z < 120. The seven PMF factors in panel **(b)** include: i) hydrocarbon-like organic aerosol (HOA), ii) BB OA (BBOA), iii) less-oxidized oxygenated OA (LO-OOA), iv) more-oxidized OOA (MO-OOA), v) less oxidized OOA associated with ammonium nitrate and biomass burning (AN-BB-OOA), vi) highly oxidized OOA associated with ammonium sulfate (AS-OOA), and vii) highly oxidized OOA associated with acidic sulfate (acidic-OOA). The dashed black line in panel **(d)** indicates f$_{60}$ = 0.3%. The elemental ratios of OA determined by the Aiken-Ambient method  (Aiken et al., 2008) are shown in the legends of panels **(e-f)**. Scatterplot of f$_{44}$ vs. f$_{60}$ **(i)** and f$_{44}$ vs. f$_{60}$ **(j)** where BB1 data are colored as a function of time of the day. The grey markers correspond to the measured OA during this study. The triangular boundaries set in panel **i** and panel **j** represent the ranges observed in ambient OOA field data from literature (Cubison et al., 2011; Ng et al., 2010)

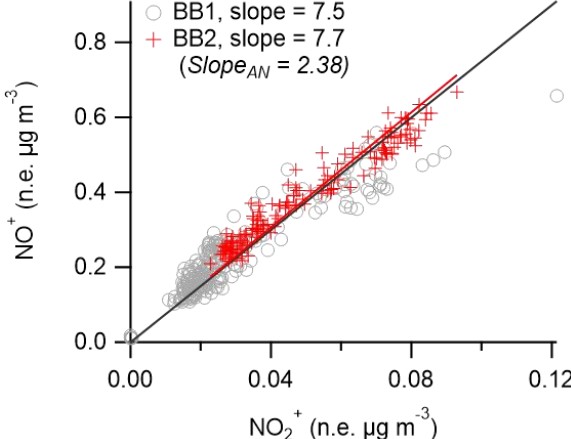

**Fig. S6.** Scatter plot between $NO^+$ and $NO_2^+$ measured by the HR-ToF-AMS during BB1 and BB2. Data fitting was performed using orthogonal distance regression.

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
