# Peer review of "Evaluation of aerosol- and gas-phase tracers for identification of transported biomass burning emissions in an industrially influenced location in Texas, USA"

_EGUsphere, 2023_

## Referee Comment (RC1)

This study evaluated aerosol and gas-phase tracers of transported biomass burning emissions in an industrially influenced location. This work has several unique elements, such as implementing an extended network of low-cost aerosol optical measurements to identify the influence of BB plumes, especially in cities designated as non-attainment or marginal nonattainment of criteria air pollutants. There are a few issues to be addressed before it can be accepted.

Major comments:

1. In your abstract, now that you highlight that both CO and acetonitrile cannot be used as a unique BB tracer for diluting BB plumes in industrially influenced locations, you ought to point out what other superior tracers are. Additionally, it is imperative to emphasize the significance and contribution of this research in this area, by explicitly stating the importance of identifying more precise and effective BB tracers for industrialized locations. This will allow readers to fully appreciate the value and relevance of the study, and make it clearer why this research is a notable and valuable addition to this field.

2. Your manuscript does not address the impacts of transported BB on urban $O_3$. Various factors such as boundary layer dynamics, transport, mixing, precursors, and local sources can complicate the observed relationship between fire influence and $O_3$ (as highlighted in references 10.1021/acs.est.2c06157 and 10.1029/2019JD031777), particularly with single-point measurements. Therefore, it would be beneficial to utilize the NOx and PTR data to provide more detailed insights into the impact of BB on $O_3$. This will greatly help to promote the impact of this manuscript.

3. I appreciate your support for the motivation behind using an extended network of low-cost aerosol optical measurements to identify the influence of BB plumes in cities designated as non-attainment or marginal non-attainment of criteria air pollutants. Nonetheless, the measurement method employed may be low in efficiency and prone to high errors. Although the authors used a combination of multiple measurement instruments, such as TAP for absorption and integrating nephelometer for scattering, they also needed to estimate the mass concentration of BC. Considering this, it is worth exploring alternative measurement instruments and methods, such as AE33 and MA200, to improve the accuracy and efficiency of the measurement process. These technologies offer advanced performance characteristics and can provide more accurate results compared to the instruments used in the present study.

4. Line 375-380, Please add the time series comparison between $NO^+$, $NO_2^+$, and AAE, or scatter plot figures, and explore the potential indication of BrC in detail.

5. Line 410-415, Why BB1 data can not be colored as a function of time of the day?

6. Section 2.2.3, Line 385-395,

In your PMF results, how did you determine and identify these factors, including less-oxidized oxygenated OA (LO-OOA), less oxidized OOA, ammonium sulfate (AS-OOA), and acidic sulfate (acidic-OOA)? These factors are not well explained or discussed in the manuscript. It will be useful to add some diagnoses for the PMF results. More discussions on the choice of PMF factors should be given.

7. Figure 6, the mobile measurement shows a significant difference between the estimated acetonitrile on drive day 1 and day 2, did the authors use the average value for the calculation of estimated acetonitrile and what was the error in the calculation?

8. PTR-MS data: It seems like the PTR-MS data are not being well leveraged to explain the temporal trends of plumes. Other VOCs like furans and phenol have been used as the BB tracer, and some carboxylic acid compounds were the main gaseous products. Do the authors consider that these species are more advantageous than acetonitrile as tracers of BB in further studies? These need to be discussed.

9. Previous field and laboratory studies have found rapid modification of aerosol and gas properties of biomass burning emissions within a few hours, such as the regional and nearfield influences of wildfire emissions (10.1021/acs.est.6b01617), the strong SOA formation and evaporation of primary semi-volatile species (10.1029/2021JD034534), change of optical properties (10.1021/acs.est.0c07569), aging effects on biomass burning aerosol mass and composition (10.1021/acs.est.9b02588). These evolutions of BB properties may influence the tracers for tracking BB sources, which may be referenced to aid some of your discussions.

Technical comments:

1. Line 233, delete the first (AAE and f60).
2. Line 46, analyzing
3. Line 55, reactions
4, Line 119, During the campaign,
5, Line 124, using Eq. (1)
6, Line 140, will result
7, Line 310, The influence of BB
8, Line 346, a significant increase
9, Line 413, an increase in $f_{44}$ and a decrease in $f_{60}$
10, Line513, can be an important factor

---

## Author Comment (AC1)

**Evaluation of aerosol- and gas-phase tracers for identification of transported biomass burning emissions in an industrially influenced location in Texas, USA**

**Sujan Shrestha et. al.**

**Corresponding author: Rebecca J. Sheesley (Rebecca_Sheesley@baylor.edu)**

The authors appreciate the comments and suggestions of the three anonymous reviewers. Detailed responses to the reviewer comments have been addressed below. Please find the reviewer's comments in black and our replies in blue. The changes in the revised manuscript are marked by track-change function. The line numbers indicated in this response correspond to the track-changed version of the revised manuscript.

**Reviewer# 1**

This study evaluated aerosol and gas-phase tracers of transported biomass burning emissions in an industrially influenced location. This work has several unique elements, such as implementing an extended network of low-cost aerosol optical measurements to identify the influence of BB plumes, especially in cities designated as non-attainment or marginal nonattainment of criteria air pollutants. There are a few issues to be addressed before it can be accepted.

Major comments:

Comment R1.1. In your abstract, now that you highlight that both CO and acetonitrile cannot be used as a unique BB tracer for diluting BB plumes in industrially influenced locations, you ought to point out what other superior tracers are. Additionally, it is imperative to emphasize the significance and contribution of this research in this area, by explicitly stating the importance of identifying more precise and effective BB tracers for industrialized locations. This will allow readers to fully appreciate the value and relevance of the study, and make it clearer why this research is a notable and valuable addition to this field.

Response R1.1. We thank the reviewer for this valuable suggestion. In response to the reviewer's comment, we have revised the abstract to emphasize the key findings of our study. Please see the revised abstract below:

"As criteria pollutants from anthropogenic emissions have declined in the US in the last two decades, biomass burning (BB) emissions are becoming more important for urban air quality.

Tracking the transported BB emissions and their impacts is challenging, especially in areas that are also burdened by anthropogenic sources like the Texas Gulf coast. During the Corpus Christi and San Antonio (CCSA) field campaign in Spring 2021, two long-range transport BB events (BB1 and BB2) were identified. The observed patterns of absorption Ångström Exponent (AAE), high-resolution time-of-flight aerosol mass spectrometer (HR-ToF-AMS) BB tracer ($f_{60}$), equivalent black carbon (eBC), acetonitrile and carbon monoxide (CO) during BB1 and BB2 indicated differences in the mixing of transported BB plumes with local anthropogenic sources. The combined information from HYSPLIT backward trajectory (BTs) and satellite observations revealed that BB1 had mixed influence of transported smoke plumes from fires in Central Mexico, the Yucatan peninsula, and the Central US, whereas BB2 was influenced majorly by fires in the Central US. The estimated transport time of smoke from the Mexican fires and the Central US fires to our study site were not too different (48-54 hours and 24-36 hours, respectively) and both events appeared to have undergone similar levels of atmospheric processing, as evident in the elemental ratios of bulk organic aerosol (OA). We observed an ageing trend for $f_{44}$ vs. $f_{60}$ and $f_{44}$ vs $f_{43}$ as a function of time during BB2, but not BB1. Positive matrix factorization (PMF) analysis of OA showed that BB1 had a mixture of organics from aged BB emission with an anthropogenic marine signal while the oxidized organic compounds from aged BB emissions dominated the aerosols during BB2. The size distribution of aerosol composition revealed distinct characteristics between BB1 and BB2, where BB1 was found to be externally mixed, exhibiting a combination of BB and anthropogenic marine aerosols. On the other hand, BB2 exhibited internal mixing, ubiquitously dominated by aged BB aerosol. Our analysis from mobile and stationary measurements highlights that both CO and acetonitrile are likely impacted by local sources even during the BB events and specifically that acetonitrile cannot be used as a unique BB tracer for dilute BB plumes in an industrially influenced location. A suitable VOC tracer would need to be emitted in high concentrations during BB, resistant to degradation during transport, unique to BB and able to be measured in the field. This study does effectively demonstrate that AAE and aerosol BB tracers served as precise and effective tracers in these complex emission scenarios. Network deployment of multiwavelength photometers holds promise for enhancing our understanding of BB impacts on air quality and supporting informed decision-making for effective mitigation strategies in locations with mixed sources and influence of dilute BB plumes. To demonstrate the relevance of such an aerosol optical network, we provide evidence of the potential regional impacts

of these transported BB events on urban $O_3$ levels using measurements from the surface air quality monitoring network in Texas."

Comment R1.2. Your manuscript does not address the impacts of transported BB on urban $O_3$. Various factors such as boundary layer dynamics, transport, mixing, precursors, and local sources can complicate the observed relationship between fire influence and $O_3$ (as highlighted in references 10.1021/acs.est.2c06157 and 10.1029/2019JD031777), particularly with single-point measurements. Therefore, it would be beneficial to utilize the $NO_x$ and PTR data to provide more detailed insights into the impact of BB on $O_3$. This will greatly help to promote the impact of this manuscript.

Response R1.2. We sincerely appreciate the reviewer's comment, and we have taken it into careful consideration. In response, we have included a new section in the manuscript, *Section 3.5: Evaluation of additional VOCs and trace gases during biomass burning events*. This section provides a brief investigation of $NO_x$ and PTR-MS data during the BB events. The added section reads as below:

"**3.5. Evaluation of additional VOCs and trace gases during biomass burning events**

Figure 7 shows the time series of acetonitrile, acetaldehyde, benzene, toluene, formaldehyde (HCHO), $O_3$, NO, $NO_2$, and $NO_x$. Notably, during BB events, benzene and toluene concentrations exhibited significant enhancement above the marine background levels, with increases of 0.23 ppbv and 0.19 ppbv (benzene), and 0.25 ppbv and 0.33 ppbv (toluene) observed for BB1 and BB2, respectively. Furthermore, elevated levels of $NO_x$ were observed during BB events compared to the marine background conditions, with enhancements of 2.81 ppbv and 2.49 ppbv noted for BB1 and BB2, respectively. Although the concentrations of these compounds were also slightly elevated above continental airmass periods, the magnitudes were comparatively lower.

Regarding acetaldehyde and HCHO, their concentrations surpassed the marine background levels; however, their profiles exhibited a photochemical production trend that correlated with the typical behavior observed for $O_3$. It should be noted that the measurement period was characterized by high ambient temperatures, which adds complexity to attributing the elevations of HCHO and acetaldehyde solely to BB plumes but does indicate enhanced photochemical activity within the BB plume. Characterizing VOCs and $NO_x$ during dilute BB plumes in an industrialized location is complicated due to the presence of multiple local emission sources and atmospheric processing

during the transport. However, the identification of the BB plumes with AAE, AMS-driven PMF factors, $f_{60}$, and satellite imagery enables attribution of these additional pollutants to the same BB plume.

In addition to acetonitrile, other VOCs like phenols, furans, furfurals, and hydrogen cyanide have also been used as BB tracers (Bruns et al., 2017; Coggon et al., 2016; Tripathi et al., 2022). However, these VOCs were not included in the select list of measured compounds. It is known that furans, furfurals, and phenols are emitted in higher quantities by biomass burning compared to vehicular emissions, making them potentially important tracers for BB plumes (Coggon et al., 2019; Mohr et al., 2013; Wang et al., 2020; Yuan et al., 2017). However, these compounds exhibit high atmospheric reactivity and undergo secondary transformations. Previous studies have demonstrated the presence of secondary oxidation products of these compounds such as maleic anhydride and nitrophenols in BB plumes (Wang et al., 2020; Yuan et al., 2017). Further, Lalchandani et al. (2022) reported a regional impact of furans from BB, leading to increased levels of SOA precursors such as ammonium nitrate and BBOA. SOA evolutions are remarkable higher during the smoldering than in flaming phase, due to the higher emission of VOCs (Li et al., 2021). Considering the high atmospheric reactivity and potential for SOA formation of these primary VOCs, further investigations into their secondary oxidized products and associated SOA are warranted in aged BB plumes. In the current study, the enhancement of MO-OOA, HCHO and acetaldehyde indicate that the plume had undergone photochemical oxidation during transport which likely would have made the measurement of reactive VOC tracers difficult."

[Figure]

Figure 7. Time series of (a) benzene and toluene; (b) NO, $NO_2$, $NO_x$ and CO; (c)acetonitrile and acetaldehyde; (d) $O_3$ and HCHO.

Comment R1.3. I appreciate your support for the motivation behind using an extended network of low-cost aerosol optical measurements to identify the influence of BB plumes in cities designated as non-attainment or marginal non-attainment of criteria air pollutants. Nonetheless, the measurement method employed may be low in efficiency and prone to high errors. Although the authors used a combination of multiple measurement instruments, such as TAP for absorption and integrating nephelometer for scattering, they also needed to estimate the mass concentration of BC. Considering this, it is worth exploring alternative measurement instruments and methods, such as AE33 and MA200, to improve the accuracy and efficiency of the measurement process. These technologies offer advanced performance characteristics and can provide more accurate results compared to the instruments used in the present study.

Response R1.3. Thank you for your valuable feedback on our manuscript. We appreciate your suggestion regarding the use of AE33 or MA200 for aerosol optical measurement instead of TAP (which is a commercial version of NOAA's continuous light absorption photometer (CLAP) (Ogren et al., 2017). We would like to take this opportunity to provide some clarification regarding our choice of instrumentation and its connection to similar monitoring networks.

Our measurement setup, including the TAP instrument, is designed to be consistent and compatible with other well-established monitoring networks such as NOAA Federated Aerosol Network (NFAN) (Andrews et al., 2019) and U.S. Department of Energy (DOE) ARM (Atmospheric Radiation Measurement) aerosol observing facility (AOS) (Flynn et al., 2018). These networks employ similar instrumentation and measurement protocols. Specially, one of the main focuses of NFAN network has been in investigating long range transport events (Denjean et al., 2016; Hallar et al., 2015). By aligning our measurement approach with these widely recognized networks, we aim to facilitate data comparison, integration, and collaboration across various research initiatives. This strategic decision allows us to contribute to a broader understanding of aerosol properties and their impacts, leveraging the wealth of information available through these established networks. Moreover, the TAP instrument utilized in this study employed a measurement wavelength of 365 nm, which is lower in the UV spectrum compared to its alternative version operating at 465 nm in the blue range. This selection allowed for measurements across a wider spectral range in the study. We acknowledge that alternative instruments such as AE33 may offer certain advantages in specific applications, especially due to their broad spectral range. We acknowledge that the narrower spectral range used by the TAP compared to AE33 may have contributed to the lower range of AAE values observed during BB events in this study. However, the protocol that we use to identify BB influence is focused on relative enhancement in the AAE above the site background as opposed to a specific literature-based threshold. This allows us to be independent of instrument-to-instrument differences in absolute absorption measurements.

While the variability in aerosol optical measurements between different instruments has been extensively studied in previous literature (Bond et al., 1999; Laing et al., 2020; Ogren, 2010; Ogren et al., 2017), it is not the primary focus of this manuscript. But we want to highlight that previous intercomparison study has demonstrated excellent agreement between long-term measurements between CLAP and particle soot absorption photometer (PSAP) at multiple sites (Ogren et al., 2017), and good agreement with the AE33 once a correction factor has been applied to the AE33 (Laing et al., 2020).

In this study, we did not consider absolute black carbon (BC) concentration as a specific tracer for BB, but rather used aerosol optical property (i.e., AAE). The AE33 and the TAP are both filter-based aerosol optical instruments, the AE33 utilizes a pre-set MAC to calculate equivalent BC. Intercomparison studies routinely demonstrate that this calculation is biased by a factor of 2-4

times above rBC. While instruments like AE33 utilize pre-set MAC values, they do not account for the site-specific variability in BC characteristics. In summary, there are significant uncertainties associated with both of these methods and co-location with a single particle soot photometer (SP2) would be the preferred methodology, which we have done in subsequent Texas studies.

In response, we have acknowledged the limitation of TAP and any filter-based absorption measurement in the Method Section 2.2.1 Aerosol optical properties (lines 159-171). The added text reads as below:

"The narrow spectral range of TAP (365nm – 540 nm) compared to other aerosol absorption measurements like aethalometer (AE33) may result in lower range of AAE. Therefore, in this study, AAE above 1.2 (i.e., average AAE during non-BB influenced period + two times standard deviation) is used to identify events that lie above the baseline AAE for a given site (discussed in Section 3.3.1), rather than absolute AAE value from the literature. While the variability in aerosol optical measurements between different instruments has been extensively studied in previous literature (ref), it is not the primary focus of this manuscript. Previous intercomparison study has demonstrated excellent agreement between long-term measurements between the CLAP (NOAA's version of TAP) and the particle soot absorption photometer (PSAP) at multiple sites (Ogren et al., 2017), and have indicated that TAP and AE33 intercompare when a different correction factor is applied to the AE33 absorption coefficient (Laing et al., 2020). The absorption coefficient data from this study is available (link below in the data availability section). This will enable future studies to access and utilize data from this study for the investigation and comparison with other instruments that uses different protocols for BC calculation."

Comment R1.4. Line 375-380, Please add the time series comparison between $NO^+$, $NO_2^+$, and AAE, or scatter plot figures, and explore the potential indication of BrC in detail.

Response R1.4. We thank the reviewer for this fruitful suggestion. In response to the reviewer's comment, we have included time series of nitrate signals attributable to organonitrates ($NO_3$-ON) in Fig 2a and added scatterplot of $NO_3$-ON vs AAE for both BB events (Fig. 5g). We have also elaborated our discussion regarding the potential indication of BrC from the ON concentration measured during this study (see below, or lines 483-503 in the revised manuscript).

"The nitrate signal measured by the AMS that are attributed to ON, referred to as $NO_3$-ON, had similar trend as AAE and $f_{60}$ during the BB events (Fig. 2a). There was a clear enhancement in $NO_3$-ON concentration during BB1 and BB2 events (0.49 ± 0.20 and 0.79 ± 0.22 µg/m$^3$, respectively) compared to the marine background period (0.21 ± 0.13 µg/m$^3$). ON is a known chromophore and thus likely contributes to the increase in AAE during BB1 and BB2. Gas phase compounds like phenols produced during BB can undergo nitrate-mediated oxidation to form aqueous phase SOA (Xiao et al., 2022). Laboratory studies have demonstrated that aqueous phase SOA from BB are chromophores and can influence the aerosol light absorption properties (Jiang et al., 2021; Pang et al., 2019; Smith et al., 2014). Additionally, BB emission can also undergo rapid oxidation by nitric radicals during nighttime to form SOA (Lalchandani et al., 2022). If we assume that the UV absorption from ON was solely responsible for enhancement in AAE during BB events, we can evaluate the relationship between AAE and ON during BB1 and BB2 (Fig. 5g). We do see a high correlation between these two during BB1 ($r^2$= 0.85) and BB2 ($r^2$= 0.95), when the correlation line is forced through zero. Thus, we observe potential indication of BrC as represented by ON during the BB events. This assumption may be an over-simplification, as other BrC compounds may also contribute to the UV absorption in these plumes."

Comment R1.5. Line 410-415, Why BB1 data can not be colored as a function of time of the day?

Response R1.5. The scatterplots of $f_{44}$ vs. $f_{43}$ and $f_{44}$ vs. $f_{60}$, where BB1 data are colored as a function of time of the day have been added to the revised SI as Figs. SI aa and bb. While the observed direction of the trend during BB2 provided information regarding photochemical ageing of the BB plume, no such distinct information regarding the aging process could be obtained during BB1. This can likely be attributed to the presence of mixed sources, including processed BB aerosols from different fire regions (discussed in *Section 3.2*) and non-BB anthropogenic emissions (discussed in *Section 3.1*), during BB1. Please see lines 457-459 for the change in the manuscript.

[Figure]

[Figure]

Comment R1.6. Section 2.2.3, Line 385-395,

In your PMF results, how did you determine and identify these factors, including less-oxidized oxygenated OA (LO-OOA), less oxidized OOA, ammonium sulfate (AS-OOA), and acidic sulfate (acidic-OOA)? These factors are not well explained or discussed in the manuscript. It will be useful to add some diagnoses for the PMF results. More discussions on the choice of PMF factors should be given.

Response R1.6. In response to the reviewer's suggestion, we have added a section describing PMF analysis on the high-resolution mass spectra in the supplementary information (see *section S2*).

"2. Positive Matrix Factorization of Organic Aerosol Matrix and Combined Organic and Inorganic Aerosol Matrices

To investigate the sources and processes of organic aerosols (OA), we performed positive matrix factorization (PMF) analysis on the high-resolution mass spectra (HRMS) of 1) organics only and 2) the combined spectral matrices of organic and inorganic species, respectively using the PMF2 algorithm in robust mode (Paatero and Tapper, 1994). We first generated the ion-speciated HRMS matrix and the corresponding error matrix from PIKA, and then analyzed using the PMF Evaluation Tool v3.06B (Ulbrich et al., 2009). We did PMF analysis on the entire sampling period covering both the stationary measurements and the mobile measurements.

For the organic PMF analysis, the OA data and error matrices were refined prior to PMF analysis according to the protocol summarized previously (Ulbrich et al., 2009; Zhang et al., 2011). Ions with m/z up to 190 were included in the PMF analysis. Isotopes were removed to avoid giving excess weight to their parent ions. Noisy ions were removed from the data matrix. These treatments

largely improved the OA factorization but had negligible impact on the mass concentrations. A minimum error was introduced for each ion. The "bad" ions with S/N ratio < 0.2 were downweighed by increasing their error values by a factor of 10, while the "weak" ions with S/N between 0.2 and 2 were downweighed a factor of 2 as described by Ulbrich et al. (2009). $O^+$, $OH^+$, $H_2O^+$, and $CO^+$ ions were also down weighted to avoid additional weight to $CO_2^+$, as their signals were all scaled to that of $CO_2^+$. PMF solutions were tested from 2 to 7 factors, and the rotational forcing parameter, fPeak, varied between -1 and 1 (step = 0.2).

We also performed PMF analysis on the combined spectral matrices of organic and inorganic species of the HR-AMS (Paatero and Tapper, 1994; Zhang et al., 2011; Zhou et al., 2017). PMF is commonly applied to the organic mass spectral matrix to determine distinct OA factors (Zhang et al., 2011). However, conducting PMF analysis on the combined spectra of organic and inorganic aerosols allows for the derivation of additional information. In this study, we performed PMF analysis on the combined HR spectral matrices of organic and inorganic species. Organic ions at m/z 12 – 180 and major inorganic ions, i.e., $SO^+$, $SO2^+$, $HSO_2^+$, $SO_3^+$, $HSO_3^+$, and $H_2SO_4^+$ for sulfate; $NO^+$ and $NO_2^+$ for nitrate; $NH^+$, $NH_2^+$, and $NH_3^+$ for ammonium; and $Cl^+$ and $HCl^+$ for chloride were included, and the ion signals were expressed in nitrate-equivalent concentrations. The error matrix was pretreated the same as the PMF analysis of organic matrix only. After PMF analysis, the mass concentration of each OA factor was derived from the sum of organic signals in the corresponding mass spectrum after applying the default RIE for organics (1.4) and the time dependent CDCE. The solutions for two to nine factors were explored at a fixed rotational parameter (FPEAK = 0). We performed similar evaluation procedures as to the organic PMF analysis and chose the seven-factor solution as the optimum solution for the combined PMF analysis. Following the procedures listed in Table 1 of (Zhang et al., 2011), all PMF solutions have been evaluated by investigating the key diagnostic plots, mass spectra, correlations with external tracers and diurnal profiles. We selected the seven-factor solution with fPeak = 0 from the PMF analysis of the combined matrices as the optimum solution. The solution is presented and discussed in detail below.

After a detailed evaluation of temporal trends, mass spectral profiles, and correlations with ions, we identified seven distinct OA factors. These seven factors are: 1) hydrocarbon-like organic aerosol (HOA) that is associated with traffic related primary emission, 2) biomass burning organic aerosol (BBOA) associated with campfires as well as regional transported wildfire plumes, 3) lessoxidized oxygenated organic aerosol (LO-OOA) representing less processed and fresher secondary organic aerosol (SOA) (O/C = 0.51), 4) more-oxidized OOA (MO-OOA) possibly representing more processed and aged SOA (O/C = 1.22), 5) an OOA that was associated with ammonium nitrate and biomass burning (AN-BB-OOA), 6) a highly oxidized OOA associated with ammonium sulfate (AS-OOA), and 7) a highly oxidized OOA associated with acidic sulfate (acidic-OOA). Three of these factors had inorganic signals in the mass spectra, and were associated with neutralized ammonium nitrate, neutralized ammonium sulfate, and acidic sulfate signals, respectively."

Comment R1.7. Figure 6, the mobile measurement shows a significant difference between the estimated acetonitrile on drive day 1 and day 2, did the authors use the average value for the calculation of estimated acetonitrile and what was the error in the calculation?

Response R1.7. We would like to provide further clarification regarding the methodology used to estimate biomass burning (BB)-related acetonitrile concentrations. The acetonitrile measurements from the mobile sampling were not directly utilized in the calculation. Instead, we adopted the enhancement ratio ($ER_{CO-acetonitrile}$ = 0.36) and background CO concentration from the study by Warneke et al. (2006). This enhancement ratio and background CO, along with the ambient CO measurements obtained in our study, was employed to estimate the contribution of BB-related acetonitrile.

To elaborate on this clarification, we have added a brief explanation in the manuscript (lines aa-aa) to explicitly state that the estimated concentrations of BB-related acetonitrile were determined using the literature-derived $ER_{CO-acetonitrile}$ and the ambient CO measurements from our investigation. Please see line 544 for the change in the revised manuscript.

Regarding the latter part of the reviewer's comment, Warneke et al. (2006) reported $ER_{CO-acetonitrile}$ of $0.36 \pm 0.06$ ppbv pptv$^{-1}$. Based on the standard deviation of the reported $ER_{CO-acetonitrile}$, we estimate that the uncertainty in our calculation for BB-related acetonitrile is ~17%. This error estimation has been included in the revised text (see lines 545-548).

Comment R1.8. PTR-MS data: It seems like the PTR-MS data are not being well leveraged to explain the temporal trends of plumes. Other VOCs like furans and phenol have been used as the BB tracer, and some carboxylic acid compounds were the main gaseous products. Do the authors

consider that these species are more advantageous than acetonitrile as tracers of BB in further studies? These need to be discussed.

Response R1.8. We thank the reviewer for this valuable suggestion. Considering this comment and earlier comment (R1.2) regarding the use of PTR-MS data and the exploration of additional biomass burning tracers, we have included a new section in the manuscript, *Section 3.5*.

Comment R1.9. Previous field and laboratory studies have found rapid modification of aerosol and gas properties of biomass burning emissions within a few hours, such as the regional and nearfield influences of wildfire emissions (10.1021/acs.est.6b01617), the strong SOA formation and evaporation of primary semi-volatile species (10.1029/2021JD034534), change of optical properties (10.1021/acs.est.0c07569), aging effects on biomass burning aerosol mass and composition (10.1021/acs.est.9b02588). These evolutions of BB properties may influence the tracers for tracking BB sources, which may be referenced to aid some of your discussions.

Response R1.9. We thank the reviewer for suggesting these references. These references have certainly helped prove our arguments in the manuscript.

Technical comments:

Comment R110. Line 233, delete the first (AAE and f60).

Response R1.11. Done

Comment R1.12. Line 46, analyzing

Response R1.12. Done

Comment R1.13. Line 55, reactions

Response R1.13. Done

Comment R1.14, Line 119, During the campaign,

Response R1.14. Done

Comment R1.15. Line 124, using Eq. (1)

Response R1.15. Done

Comment R1.16. Line 140, will result

Response R1.16. Done

Comment R1.17. Line 310, The influence of BB

Response R1.17. Done

Comment R1.18. Line 346, a significant increase

Response R1.18. Done

Comment R1.19. Line 413, an increase in $f_{44}$ and a decrease in $f_{60}$

Response R1.19. Done

Comment R1.20. Line513, can be an important factor

Response R1.20. Done

**Reviewer 2**

**General comment**

Comment R2.1. The paper is focused on the characterization of biomass burning events impacting SW Texas, and their association with local air quality, while trying to disentangle the impact of urban and regional anthropogenic sources. The subject is timely, and it is treated in an interesting multi-angle approach combining high-end instrumentation (AMS, PTR MS) with mobile measurements and low-cost photometers.

The inadequacy of acetonitrile as a BB tracer in urban environments is an important result, well-justified by this study. However, a similar result for CO should have been expected and probably its description as a salient finding could be toned down. In the absence of clear markers for aged BB, the combination of AMS-driven PMF analysis backed up by satellite imagery and trajectory analysis appears as a key option for the characterization of processed aerosol from wildfires. Although this might not be feasible everywhere, it is a main message of the manuscript and should be stressed further.

This study also verifies that measurements and analysis of optical properties (although not without limitations) provide also possibilities for BB aerosol identification (also if Brown Carbon aerosol can be apportioned), but advanced multi-wavelength photometers are necessary to this end in order to reduce uncertainties. The used absorption photometers provide valuable solutions for flexible

monitoring, but there are some inherent biases that should be acknowledged, as most probably there would be more confidence in the absorption results if a desktop multi-wavelength photometer had been available.

Overall the paper is well-written, well-referenced and it can be considered for publication after sharpening its take-home messages based on the arguments above, and addressing the following technical comments.

Response R2.1. We would like to express our sincere gratitude to the reviewer for acknowledging the significance and relevance of our manuscript. Your insightful suggestions have greatly contributed to improving the scientific quality of the manuscript.

We would like to acknowledge the reviewer for raising the point regarding toning down the emphasis on the CO results in our study. We agree that the sources of CO, including anthropogenic and biomass burning, have been extensively studied and reported in the literature. Therefore, we have taken this feedback into consideration and made appropriate revisions to the abstract to provide a more balanced representation of the salient features of our study.

In regard to the latter comment regarding the choice of instrumentation for aerosol measurement and the biases in measurement, Reviewer #1 also had a similar comment. So, refer to Comment R1.3 for our response. In that response, we provide a detailed justification for our instrumentation selection and address any concerns regarding measurement biases.

**Specific comments**

Comment R2.4. Abstract: Some results related to the mass size distributions and different mixing states of BB1, BB2 should be included in the abstract.

Response R2.4. We agree with the reviewer's comments. The results related to NR-PM$_1$ mass size distribution and different mixing state of BB1 and BB2 are added to manuscript. The added text reads as below:

The size distribution of aerosol composition revealed distinct characteristics between BB1 and BB2, where BB1 was found to be externally mixed, exhibiting a combination of BB and anthropogenic marine aerosols. On the other hand, BB2 exhibited internal mixing, ubiquitously dominated by aged BB aerosol.

Comment R2.5. Line 47: It is somewhat of a stretch to classify CO, BC and even BrC as BB markers. It would be better to rephrase.

Response R2.5. We have rephrased the statement, following the reviewer's suggestion (see lines 57-58).

Comment R2.6. Line 52: SSA on its own does not characterize wavelength dependence.

Response R2.6. We agree with the reviewer that SSA is rather used to provides information about the top-of-atmosphere forcing due to aerosol than wavelength dependence of aerosol. So, we have edited the text in the revised manuscript.

Comment R2.7. Line 61: Changes on episodic scale or affecting air quality indicators in the long run?

Response R2.7. In recent decades, the Western United States has experienced a notable increase in forest wildfire activity. This escalation is characterized by higher frequencies of large wildfires for longer durations, and extended wildfire seasons. Consequently, downwind locations have witnessed elevated concentrations of air pollutants during the wildfire season, which typically occurs in spring and summer.

We have revised the text to highlight the increase in air pollutants in the downwind locations during wildfire seasons due to these fires. Please see line 71 for the change in the manuscript.

Comment R2.8. Line 63: Again, which is the temporal scale of this exacerbation?

Response R2.8. The references included in the manuscript has shown that the Alaskan, Canadian and US pacific northwest fires transported to Houston and exacerbated $O_3$, $PM_{2.5}$, CO, and BC concentrations for several days. We have revised the text in the manuscript accordingly (see line 74).

Comment R2.9. Lines 59-64: Impacts from agricultural burning have also been extensively documented in the SE US.

Response R2.9. We thank the reviewer for pointing this out. We have added references for the studies related to forest fires and agricultural burnings in the southeastern US. Please see lines 77-79.

Comment R2.10. Line 70: Not clear how BB transport will lead into ozone exceedances. Please explain, for the specific case. It should be also considered that BB plumes containing absorbing BC and organics could also modulate photolysis and have a reverse O3 effect than the one described.

Response R2.10. We thank the reviewer for this suggestion. In response, we have included possible reasons that lead to $O_3$ exceedances due to transported BB plumes in the downwind regions. Further, we have acknowledged that carbonaceous aerosols and organics can absorb and scatter incoming solar radiation and alter the photolysis of atmospheric trace gases, thereby reducing $O_3$ production. Please see lines 80-86 for the change in the manuscript.

Comment R2.11. Lines 83-84: Mention the frequency.

Response R2.11. Based on historical wind data, it has been observed that during the spring month in San Antonio, the prevailing wind direction is predominantly southeasterly (Guo et al., 2021). We have incorporated this information into the revised text to accurately reflect the wind patterns in the region (see lines 105-106).

Comment R2.12. Lines 136-137: It would be better to keep the "identify events above the baseline" and omit the "indicate periods of BB influence", since based on results from aerosol typing studies an AAE of 1.2 might be too low to indicate pure BB aerosol.

Response R2.12. We agree with the reviewer's suggestion. The text in the manuscript has been revised accordingly (see lines 161-162).

Comment R2.13. Line 140-142: Take into account that these estimates refer to top-of-atmosphere forcing.

Response R2.13. We agree with the reviewer's suggestion. The text in the manuscript has been revised accordingly (see line 173).

Comment R2.14. Line 152: Mention also here the MAC value you calculated. You should also acknowledge limitations around loading and multi-scattering absorption effects that are not compensated in the TAP/CLAP.

Response R2.14. Using the method discussed in the manuscript, the derived MAC at 520 nm was 11.45 $m^2g^{-1}$. We have included this information in the revised manuscript (see line 185).

Regarding the latter comment, we have acknowledged the limitations of using TAP in aerosol absorption measurement. We want to clarify to the reviewer that filter loading effect is corrected in the TAP data but scattering correction has not been applied (see line 159-171).

Comment R2.15. Line 158-165: Are these results from PMF analysis conducted in the present paper? More details are needed.

Response R2.15. The PMF analysis was performed using the entire dataset collected during the campaign, and a subset of the results was utilized to interpret the biomass burning events identified in this study. Following the reviewer's suggestion, we included additional details of the PMF analysis in the *Supplemental Information S2*.

Comment R2.16. Section 2.1: The monitoring periods should be defined here.

Response R2.16. Thanks for the suggestion. We have included the measurement period at Port Aransas/ Corpus Christi in the revised manuscript (see line 117).

Comment R2.17. Line 253-254: The AAE values for the events are somewhat low, for what is usually expected for BB aerosols. Do you expect that uncertainties in absorption measurements by the low-cost devices in the near-UV range play a part in this?

Response R2.17. Reviewer 1 also had similar concerns. Please refer to Comment R1.3 for our response.

Comment R2.18. Line 284: Would this comment imply that UV-absorbing chromophore would be more susceptible to photo-bleaching? It tends to be the other way round (non-polar chromophores absorbing in the visible range tend to be more sensitive to degradation). Discuss.

Response R2.18. In this referenced line, we aimed to indicate that photobleaching of BrC and boundary layer mixing of BB plumes with anthropogenic emissions may have subsequently decreased the AAE during the transport. However, we acknowledge that BrC ageing is a complex process. Further, this study did not have plume transect data capturing the BB plume transport, to specifically characterize the effect of photobleaching on the specific type of chromophores.

Comment R2.19. Section 3.3: It is not clear how shipping emissions translate into the PMF factors identified here. Based on recent literature, there is the possibility to both influence HOA from near-coast activity, and OOA factors as processed aerosol from open-sea navigation. Some implications

should be included here, since due to the location of the measurements, such activity can drive non-BB OA.

Response R2.19. We agree with the reviewer regarding the possibility of marine anthropogenic activities on non-BB OA factors due to the nature of the sampling location in this study. The recent work from the same authors of this manuscript found that during the sampling period, marine "background" aerosols from the Gulf of Mexico were greatly influenced by anthropogenic activities, likely processed shipping emissions over the Gulf (Zhou et al., 2023); sulfate and two OOA factors, AS-OOA and acidic-OOA, were elevated significantly. Figure S5 shows that the aerosol (NR-PM$_1$) composition during these periods were dominated by sulfate, and the OA composition was dominated by AS-OOA while acidic-OOA were also elevated during the marine background periods, suggestive of influences from processed shipping emissions. During the BB periods, however, organics became the dominant NR-PM$_1$ component and the mass fractional contribution of SO$_4$ decreased dramatically while BBOA and MO-OOA increased. This suggests that the OA composition during the BB periods were predominantly driven by biomass burning plumes; marine anthropogenic activities likely had minimal influence on the non-BB OAs. Please refer to lines 418-423 for the change in the manuscript.

The same authors also observed episodic brief but intensive positive perturbations in the NR-PM$_1$ time series due to intense primary aerosol emissions from near-coast activities including very local vehicular exhaust and on-the-beach campfires and cooking. Signature ions for HOA (e.g., C$_4$H$_9^+$) and BBOA (e.g., C$_2$H$_4$O$_2^+$) showed spiky behavior during these events. As shown in Figure S3 of Zhou et al. (2023), most of these spiky near-coast activities yielded organic concentrations reaching above 40 ug/m$^3$, higher than those during the BB events, and took place during the first week of the sampling period, prior to the BB event days.

Comment R2.20. Line 348: It is difficult to follow which is the study period you are referring to. Is it 3-15 April, or just the event days.

Response R2.20. We appreciate the opportunity to clarify this point. In this study, unless otherwise specified, all the data presented pertains to the period of interest, i.e., 9 – 11 April. We have clarified regarding this in the beginning of the results and discussion section (see lines 275).

Comment R2.21. Line 350-352: Or it could be interpreted as the BB events not being severe enough to have an impact at ground level.

Response R2.21. We agree with the reviewer and have acknowledged this argument in the revised manuscript (see line 393)

Comment R2.22. Line 362: What do you mean by lower range? Your reported fractions place somewhere in the middle of the values from the other listed studies.

Response R2.22. We thank the reviewer for pointing this out. We have revised the text to reflect moderate range OA fraction observed during the BB events. Please see line 404 for the change in the manuscript.

Comment R2.23. Line 364: Write correctly the chemical formula for the sulfate ion.

Response R2.23. Done

Comment R2.24. Line 395: Rephrase, it cannot be supported that MO-OOA factors in AMS studies are generally indicators of aged BB.

Response R2.24. We thank the reviewer for this suggestion. We have rephrased the text in the manuscript (see lines 435).

Comment R2.25. Figure 7: Event periods could be shaded similar to the other line figures.

Response R2.25. We thank the reviewer for this suggestion. We have revised this figure with BB1 and BB2 shaded like the previous time series figures.

For editing

Comment R2.26. Line 26: "as a function of time elapsed during BB2". Not clear.

Response R2.26. Edited as "We observed an ageing trend for $f_{44}$ vs. $f_{60}$ and $f_{44}$ vs. $f_{43}$ as a function of time during BB2..."

Comment R2.27. Line 65: Verb missing

Response R2.27. Revised as "Wildfires and agricultural burning in the Central Mexico and the Yucatan peninsula peak during the spring-summer season and also transport pollutants to the Southern US".

Comment R2.28. Line 90: "anthropogenic influenced area". Simplify.

Response R2.28. Revised as "industrial area"

Comment R2.29. Lines 171-173: Rewrite sentence, difficult to read.

Response R2.29. Revised as: "O$_3$ measurements were conducted using a modified Thermo Environmental, Inc., Model 42C instrument, which utilizes chemiluminescence (CL) with NO gas to measure O$_3$."

Comment R2.30. Line 217: "Results".

Response R2.30. Done

[revised manuscript text omitted]